# Role of Rabenosyn-5 and Rab5b in host cell cytosol uptake reveals conservation of endosomal transport in malaria parasites

Ricarda Sabitzki[1], Anna-Lena Roßmann[1], Marius Schmitt[1], Sven Flemming[1], Andrés Guillén-Samander[1], Hannah Michaela Behrens[1], Ernst Jonscher[1], Katharina Höhn[2], Ulrike Fröhlke[1], Tobias Spielmann[1]*

1 Pathogen Section, Bernhard Nocht Institute for Tropical Medicine, Hamburg, Germany, 2 Electron Microscopy Unit, Bernhard Nocht Institute for Tropical Medicine, Hamburg, Germany

* spielmann@bnitm.de

**Data Availability Statement:** All relevant data are within the paper and its Supporting Information files.

## Abstract

Vesicular trafficking, including secretion and endocytosis, plays fundamental roles in the unique biology of *Plasmodium falciparum* blood-stage parasites. Endocytosis of host cell cytosol (HCC) provides nutrients and room for parasite growth and is critical for the action of antimalarial drugs and parasite drug resistance. Previous work showed that PfVPS45 functions in endosomal transport of HCC to the parasite's food vacuole, raising the possibility that malaria parasites possess a canonical endolysosomal system. However, the seeming absence of VPS45-typical functional interactors such as rabenosyn 5 (Rbsn5) and the repurposing of Rab5 isoforms and other endolysosomal proteins for secretion in apicomplexans question this idea. Here, we identified a parasite Rbsn5-like protein and show that it functions with VPS45 in the endosomal transport of HCC. We also show that PfRab5b but not PfRab5a is involved in the same process. Inactivation of PfRbsn5L resulted in PI3P and PfRab5b decorated HCC-filled vesicles, typical for endosomal compartments. Overall, this indicates that despite the low sequence conservation of PfRbsn5L and the unusual N-terminal modification of PfRab5b, principles of endosomal transport in malaria parasite are similar to that of model organisms. Using a conditional double protein inactivation system, we further provide evidence that the PfKelch13 compartment, an unusual apicomplexa-specific endocytosis structure at the parasite plasma membrane, is connected upstream of the Rbsn5L/VPS45/Rab5b-dependent endosomal route. Altogether, this work indicates that HCC uptake consists of a highly parasite-specific part that feeds endocytosed material into an endosomal system containing more canonical elements, leading to the delivery of HCC to the food vacuole.

## Introduction

Malaria caused by *Plasmodium falciparum* parasites remains an important cause of infectious disease-related death [1]. The pathology of malaria is caused by the asexual development of the

**Funding:** This work was funded by the European Research Council (ERC, grant 101021493 to TS) and the DFG funded research training group GRK2771 (to TS). MS thanks for funding from the Jürgen Manchot Stiftung and AGS thanks for funding from EMBO (Postdoctoral Fellowship EMBO ALTF Number 166-2022). The funders had no role in study design, data collection and analysis, decision to publish, or preparation of the manuscript.

**Competing interests:** The authors have declared that no competing interests exist.

**Abbreviations:** CoIP, co-immunoprecipitation; DIC, differential interference contrast; ELC, endosome-like compartment; FCP, FYVE-containing protein; HCC, host cell cytosol; HCCU, host cell cytosol uptake; IFA, immunofluorescence assay; NLS, nuclear localization signal; PAGE, polyacrylamide gel electrophoresis; PI3P, phosphatidylinositol 3-phosphate; POI, protein of interest; PPM, parasite plasma membrane; PVM, parasitophorous vacuolar membrane; RBC, red blood cell; SLI, selection linked integration.

parasite within red blood cells (RBCs) of the host. Progress in reducing the global impact of malaria has slowed recently and drug resistance was identified as one factor jeopardizing malaria control [1]. The action of several antimalarial drugs critically depends on the degradation of host cell hemoglobin in the parasite's lysosome-like compartment termed the food vacuole [2]. The hemoglobin derives from cytosol the parasite takes up from the host cell in an endocytic process. Hemoglobin degradation products activate the current first-line drug artemisinin (and derivatives), and decreased susceptibility to these drugs is associated with a reduced host cell cytosol uptake (HCCU) [3,4]. The endocytosed hemoglobin is a source of amino acids for the parasite [5]. Consequently, amino acid availability is a growth restricting factor in parasites with a reduced susceptibility to artemisinin, indicating a trade-off between artemisinin susceptibility and HCCU levels [6]. HCCU is also critical for providing space for parasite growth and for the osmotic stability of the infected host cell [7]. The uptake and digestion of hemoglobin hence constitute a vulnerability for the parasite. However, despite its importance, the molecular basis for HCCU is not well understood [8].

Compared to eukaryotic model organisms, endocytosis in malaria blood stages faces particular challenges arising from the unique environment in which the intracellular parasite resides [8]. The parasites develop surrounded by a milieu of high protein density (mainly hemoglobin), from which they are separated not only by their plasma membrane (PPM) but also by an additional membrane, the parasitophorous vacuolar membrane (PVM). Morphological studies implicated the cytostome, an invagination of the PPM and the surrounding PVM, as the site where HCCU is initiated at the PPM [9–12]. However, the mechanism of how endocytic structures are formed remains unclear [8], and only recently functional data directly implicated specific parasite proteins in this process [3,13–15]. Most of these proteins are located at an electron dense collar surrounding the cytostomal neck and are involved in the early phase of HCCU (i.e., for the presumed initiation and formation of endocytic vesicular structures at the PPM) [3,4,15,16]. The majority of the proteins at the cytostomal collar do not resemble typical endocytosis proteins [3,15], indicating that the initiation of endocytosis for HCCU displays strong parasite-specific adaptations that—based on recent work—are conserved in apicomplexans [17,18].

Less is known about proteins in later phases of HCCU, the transport of internalized HCC to the parasite food vacuole. Inactivation of the parasite's orthologue of VPS45 leads to an accumulation of HCC-filled vesicles in the parasite [13]. This indicated that PfVPS45 is involved in HCC transport, resembling the function of its orthologues in model organisms that are needed for the transport of endosomal cargo to the lysosome [19,20]. The HCC-filled vesicles induced after PfVPS45 inactivation are enclosed by 1 or 2 membranes, can contain smaller internal vesicles similar to endosomes in model organisms, and often harbor phosphatidylinositol 3-phosphate (PI3P) in their membrane facing the cytosol, overall suggesting endosomal characteristics [13].

The presence and function of VPS45 in the parasite may indicate that endosomal transport in the parasite follows a more canonical pathway than endocytosis initiation at the PPM. PfVPS45, PI3P kinase, the phosphoinositide-binding protein PfPX1, host Peroxiredoxin-6, and actin [13,14,21–23] have so far been implicated in endosomal transport of HCC. However, the identification of proteins involved in this process is difficult because in apicomplexan parasites, many homologs of endolysosomal proteins appear to have been repurposed for functions associated with the specialized secretory organelles needed for host cell invasion [24–29]. Hence, similarity to endolysosomal proteins has limited predictability to identify such proteins in malaria parasites. It is for instance still unclear whether PfRab5 isoforms are involved in HCCU. *P. falciparum* Rab5a, initially thought to function in HCCU [10], is only essential in schizonts, not trophozoites, indicating no role in HCCU [30]. PfRab5b has been shown to

localize to the parasite food vacuole and the PPM [31,32]. However, direct functional data is lacking for both PfRab5b and PfRab5c and it is at present unknown if they are involved in HCCU or not. In model organisms, VPS45 typically functions together with Rab5 and the Rab5-effector rabenosyn5 (Rbsn5) (in mammals) or Vac1/PEP7 (in yeast) [33–35] in a fusion complex important for endosome maturation [33]. However, if and which Rab5 is involved in HCCU in malaria parasites is unknown and a Rbsn5 has not been detected in the parasite's genome.

Here, we identified a *P. falciparum* Rbsn5-like (PfRbsn5L) protein and showed that it interacts and functions with PfVPS45 and PfRab5b in the transport of HCC to the food vacuole. Our data provide evidence that the *P. falciparum* Rab5-Rbsn5-VPS45 fusion complex—and thus elements of this part of the endosomal pathway—is evolutionarily conserved although the binding specificity of the PfRbsn5L FYVE domain remains unknown. Additionally, double conditional inactivation of PfRbsn5L together with a cytostomal collar protein involved in the early phase of endocytosis at the PPM suggests that the HCC-containing vesicles originated from the cytostome. Overall, our data suggest that HCCU consists of a parasite-specific initial part at the PPM that delivers endocytosed material into an endosomal system that contains more canonical aspects.

## Results

### Identification of a putative *P. falciparum* Rbsn5-like protein

If PfVPS45-dependent endosomal transport is evolutionarily conserved, it is expected to also depend on an equivalent of Rbsn5 or Vac1/PEP7 which up to now had been elusive in malaria parasites. In order to identify possible Rbsn5 candidates, we conducted in silico searches. Rbsn5 from other organisms contain an FYVE-type zinc finger but only a single FYVE-domain containing protein was previously detected in the *P. falciparum* genome and named FYVE-containing protein (FCP) [36]. BLAST searches using human Rbsn5 (Q9H1K0) identified FCP as the top hit. However, the BLAST-detected similarity (46% identity) was restricted to 37 amino acids of the FYVE domain (score 51.2; E value 4e-07). It was therefore unclear whether FCP corresponds to the *P. falciparum* Rbsn5 or to a different FYVE-domain protein such as, e.g., the early endosomal antigen 1 (EEA1). To clarify this, we used HHPred [37,38] to query the *P. falciparum* proteome with human Rbsn5 which identified a different protein, PF3D7_1310300, as the top hit. This protein displayed similarity to HsRbsn5 over 212 (E value 5e-12) of its 247 amino acids and contains a FYVE/PHD zinc finger in its N-terminal half (Figs 1A and S1A). As the HHPred detected similarity went beyond the FYVE domain, we reasoned that this was the most likely candidate for PfRbsn5. Alignment of PF3D7_1310300 with HsRbsn5 showed that the *P. falciparum* protein missed the region containing the N-terminal C2H2 zinc finger and the NPF repeat region in the C-terminal half of human Rbsn5 (Fig 1A) but displayed 52.2% similarity over its entire sequence with the corresponding region of HsRbsn5, whereas FCP showed 43.4% similarity over its entire sequence with the corresponding region of HsRbsn5. PF3D7_1310300 was also the top hit if the HHPred search was repeated using yeast VAC1/PEP7 (E-value 8.2e-12) which is considered the likely single equivalents of Rbsn5 and EEA1 in yeast [33]. FCP was the second-best hit of this HHPred search but covered less sequence (E value 3.3e-10). Nevertheless, FCP and PF3D7_1310300 show only a low level of conservation between themselves. As malaria parasites contain 2 FYVE proteins with similarities to VAC1/PEP7 (FCP and PF3D7_1310300), we followed the nomenclature for mammalian cells and tentatively named the one with the higher similarity and longer region of similarity to human Rbsn5 (PF3D7_1310300) *P. falciparum* Rabenosyn-5-like protein (PfRbsn5L). Besides the FYVE domain, PfRbsn5L also contained a region with some

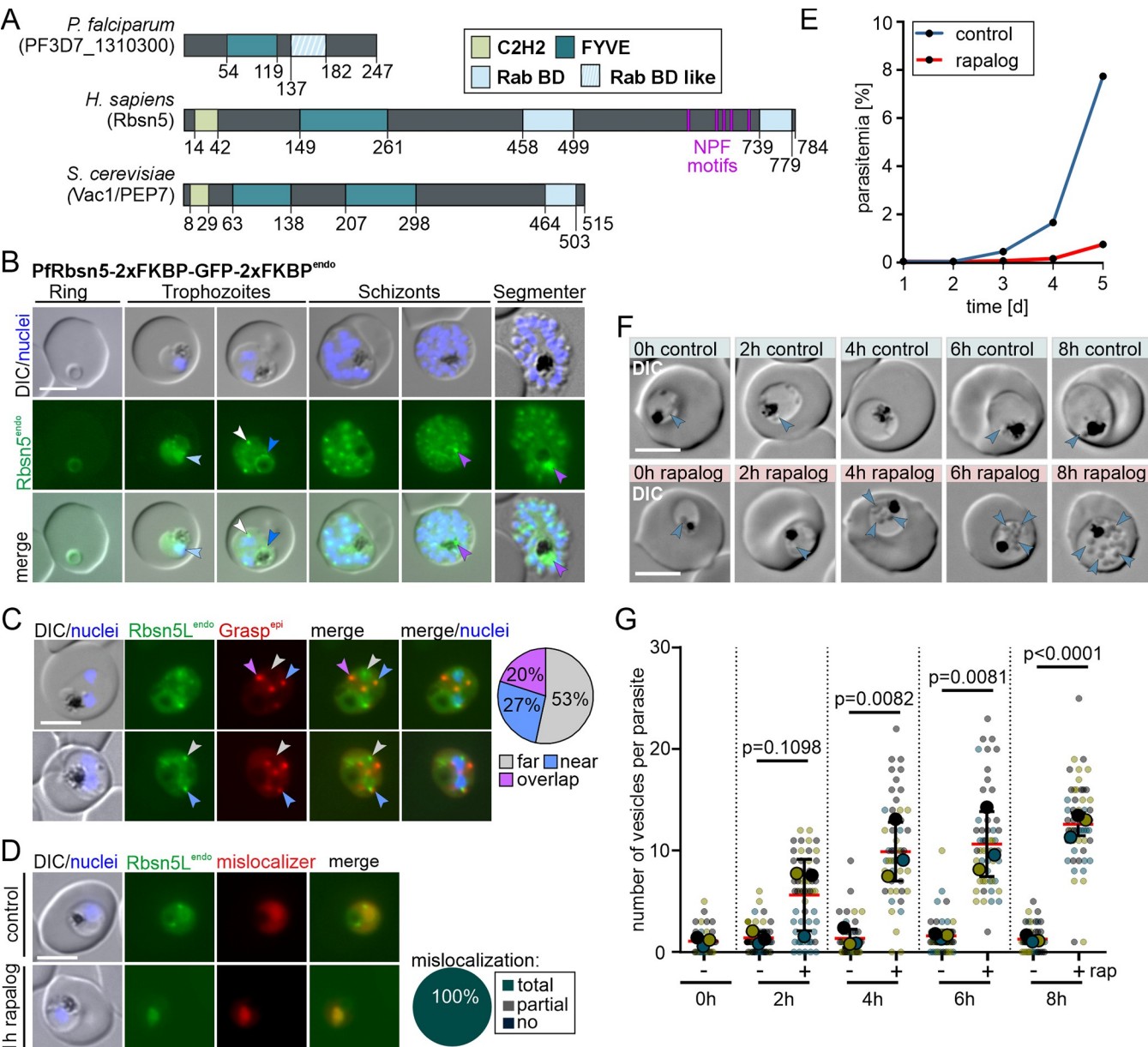

**Fig 1. Conditional Inactivation of PfRbsn5L leads to vesicles in the parasite and parasite death. (A)** Comparison of domain architecture of the putative PfRbsn5L, *H. sapiens* HsRbsn5, and *S. cerevisiae* VAC1/PEP7. BD, binding domain C2H2. **(B)** Live-cell microscopy images of the indicated stages of PfRbsn5L-2xFKBP-GFP-2xFKBP[endo] parasites (white arrow, nucleus-proximal PfRbsn5L[endo] foci; light blue arrow, faint dispersed signal in the nucleus; dark blue arrow, signal at the food vacuole; purple arrow, intense focus at the food vacuole in schizont stage parasites). **(C)** Live-cell microscopy images of PfRbsn5L-2xFKBP-GFP-2xFKBP[endo] parasites, co-expressing Grasp[epi]. Arrows show PfRbsn5L[endo] foci close to nuclei and are color coded based on overlap with the closest GRASP focus. Overlap: full overlap; near: less than GRASP focus diameter apart, far: more than one focus diameter apart. Pie shows proportion of these foci (*n* = 45 with cells from 3 independent imaging sessions). **(D)** Live-cell microscopy images of PfRbsn5L (PfRbsn5L-2xFKBP-GFP-2xFKBP[endo]+ *nmd3*'NLS-FRB-mCh[epi] parasites) using a nuclear mislocalizer (mislocalizer) [30] 1 h after induction of knock-sideway (1 h rapalog) compared to the control (control). Knock-sideway was classified (pie chart) as complete (no signal detected outside the nucleus), partial (signal in the nucleus but also at original site), or absent (no) mislocalization in *n* = 30 parasites from 2 independent experiments. **(E)** Flow cytometry-based growth curve over 2.5 growth cycles of PfRbsn5L knock-sideways (rapalog) compared to the control parasites. One representative of *n* = 3 independent experiments, all replicas shown in (S1F Fig). **(F)** Representative DIC live-cell images of parasites 0 h, 2 h, 6 h, and 8 h after induction of knock-sideways of PfRbsn5L (+ rapalog) compared to control. Blue arrows, vesicular structures. **(G)** Quantification of number of vesicles in synchronous trophozoites 0 h, 2 h, 6 h, and 8 h after induction of PfRbsn5L knock-sideways. Data shown as superplot [77] from *n* = 3 independent experiments (individual experiments: blue, yellow, and black with 147, 176, and 144 parasites (small dots), respectively; average of each experiment as large dot); two-tailed unpaired *t* test; red lines, mean; black lines, error bar (SD); *p*-values indicated. Scale bars, 5 μm and 1 μm in the magnifications. Nuclei were stained with DAPI. DIC, differential interference contrast; endo, endogenous; epi, episomal; Rbsn5L[endo], 2xFKBP-GFP-2xFKBP-tagged Rbsn5L expressed from endogenous locus. The data underlying this figure can be found in S1 Data.

similarity to the Rab5-binding domain present in other Rabenosyn5s (S1B Fig). Interestingly, a closer inspection of the amino acid sequence of the PfRbsn5L FYVE domain revealed differences to conserved PI3P binding residues although the general fold of its AlphaFold2 predicted structure [39,40] closely matched experimental FYVE domain structures (S1A and S1C Fig). The amino acid positions deviating from the consensus in the PfRbsn5L FYVE were the same that also deviated in the FYVE domain of human protrudin which—in contrast to other FYVE domains—does not bind PI3P but other phosphoinositides [41]. In contrast, the FYVE domain of FCP matched the consensus (S1A Fig). We expressed a tandem of the PfRbsn5L FYVE domain fused to mCherry in parasite also expressing the PI3P reporter P40X fused to GFP [42]. The 2xFYVE domain construct appeared uniformly distributed in the parasite cytoplasm and we did not find any accumulation at PI3P containing structures (S1D Fig). We conclude that the PfRbsn5L FYVE domain does not bind PI3P or other membrane lipids. Alternatively, it might bind rare phosphoinositide species that are not sufficiently abundant to lead to a recruitment detectable above background or if requires cooperativity with other regions or interactors of PfRbsn5L to mediate binding.

## Subcellular localization and conditional inactivation of PfRbsn5L

To investigate the localization and function of PfRbsn5L, we tagged the *rbsn5l* gene with the sequence encoding 2xFKBP-GFP-2xFKBP using the selection linked integration (SLI) system to modify the endogenous locus [30]. The resulting PfRbsn5L-2xFKBP-GFP-2xFKBP[endo] cell line (short PfRbsn5L[endo]) (Figs 1B and S2A) showed PfRbsn5L in foci and accumulations in addition to a general cytosolic distribution. The most prominent accumulations were foci in proximity to the nucleus (Fig 1B, white arrow) that increased in number with the increasing number of nuclei during progression of the parasite blood cycle. In addition, a signal was present at the food vacuole in trophozoite stages (Fig 1B, dark blue arrow) with a more intense accumulation in proximity of the food vacuole in schizont stage parasites (Fig 1B, purple arrows). In some cells, a faint dispersed signal overlapping with the DAPI-stained nuclei was observed (Fig 1B, light blue arrow). Co-expression of a fluorescently tagged Grasp[epi] showed that only about half of the PfRbsn5L foci at the nuclei were in close proximity or overlapped with the Golgi-apparatus (Figs 1C, white arrows and S2B) and apart from the nucleus proximal foci, PfRbsn5L foci did not regularly overlap with the ER (S2C Fig). Overall, the localization of PfRbsn5L was similar to the one we previously observed for PfVPS45 in the PfVPS45-2xFKBP-GFP[endo] cell line [13].

To investigate its function, we conditionally inactivated the parasite's Rbsn5L using knocksideways [30,43,44], an approach particularly suited for rapid inactivation of target proteins [45]. This method is based on the FRB-FKBP dimerization system. One domain of this system is fused to the protein of interest (POI) and the other domain is fused to a trafficking signal, e.g., a nuclear localization signal (NLS) (the so-called mislocalizer). Upon addition of a small ligand (rapalog), the POI and the mislocalizer dimerize and by virtue of the trafficking signal on the mislocalizer, the POI is removed from its site of action, e.g., to the nucleus if an NLS is used. We episomally expressed the mislocalizer (*nmd3*'1xNLS-FRB-mCh[epi]) in the PfRbsn5L-2xFKBP-GFP-2xFKBP[endo] cell line. Upon addition of rapalog, PfRbsn5L was efficiently mislocalized to the nucleus within 1 h (Fig 1D). To determine the relevance of PfRbsn5L for parasite blood stage development, we monitored the parasitemia in cultures grown in presence or absence (control) of rapalog over 5 days using flow cytometry. PfRbsn5L inactivation led to a substantial growth defect in comparison to the control parasites (Figs 1E and S2D), indicating an important function of PfRbsn5L for the asexual blood stages of *P. falciparum* parasites. Assessment of growth in synchronous parasites showed that inactivation of PfRbsn5L in ring

stages prevented development into trophozoites in the majority of parasites, whereas inactivation at the transition to the trophozoite stage resulted in a marked accumulation of aberrant late stage parasites of which most failed to give rise to new rings (S3 Fig).

## Inactivation of Rbsn5L leads to accumulation of vesicles with endosomal characteristics

Monitoring the parasites in a narrower time frame via differential interference contrast (DIC) microscopy showed that the inactivation of PfRbsn5L led to an accumulation of vesicular structures in the parasite cell (Fig 1F, blue arrows), a phenotype previously observed upon PfVPS45 inactivation [13]. The number of these vesicles per parasite increased over time from an average of 1.06 ± 0.25 at induction of the knock-sideway (+) to an average of 5.62 ± 2.03 two hours and 12.59 ± 0.65 eight hours after PfRbsn5L inactivation (Fig 1F, blue arrows and 1G), while the number of vesicles stayed low in control (-) parasites (1.26 ± 0.21) (Fig 1F, white arrows and 1G). The diameter of the parasites with the inactivated PfRbsn5L showed no significant difference to the control at the respective time points (S4A Fig), indicating that the phenotype was not due to a loss of parasite viability during the experimental period.

To investigate the observed phenotype in more detail, we analyzed the parasites by electron microscopy. Whereas vesicular structures were only occasionally observed in the control parasites, multiple vesicular structures were present in the parasites with inactivated PfRbsn5L (Fig 2A). These vesicular structures contained material with a similar electron density to that of the host cell cytosol (Fig 2A) and hence likely represent structures of HCC internalization. Additionally, some of the vesicular structures contained vesicles of smaller dimensions (Fig 2A, white arrows), a detail we also observed before in vesicles induced upon PfVPS45 inactivation [13] and that might correspond to intraluminal bodies found in endosomes of model organisms.

In order to test whether the observed vesicular structures could be intermediates of HCCU, we performed bloated food vacuole assays [3,13]. For this assay, the parasites were treated with the protease inhibitor E64, which prevents the digestion of hemoglobin in the food vacuole [46]. As a consequence, newly internalized hemoglobin reaching the food vacuole accumulates, resulting in a bloated food vacuole phenotype if HCCU is operational. While nearly all control parasites showed a bloated food vacuole, the food vacuole in the PfRbsn5L inactivated parasites did not bloat (Fig 2B). This finding indicated an impairment of the HCC delivery pathway and demonstrates the importance of PfRbsn5L function in this process. This effect was not due to a loss of parasite viability, as the diameter of the parasites with inactivated PfRbsn5L showed no significant difference to the control over the assay time (S4B Fig). To further confirm the specificity of this effect, we conducted a bloated food vacuole assay after inactivation of the essential vesicle trafficking protein PfSand1 which is likely not involved in HCCU [30]. In contrast to the inactivation of PfRbsn5L, inactivation of PfSand1 (using the PfSand1-2xFKBP-GFP[endo] + NLS-mislocalizer[epi] cell line) did not prevent bloating of the food vacuole, equivalent to the untreated control (Fig 2C).

We also carried out electron microscopy examinations of the cells that had completed the bloated food vacuole assay: after inactivation of PfRbsn5L the parasites showed small food vacuoles that were much less electron-dense than the host cell and vesicular structures in the cytoplasm with electron-dense material, whereas controls showed an enlarged food vacuole filled with electron-dense material (Fig 2D).

In order to assess whether the vesicle accumulation and endocytosis defect cause the viability loss of the parasites after PfRbsn5L inactivation, we inspected DIC images taken from the growth experiments with synchronous parasites (S3 Fig). At the time points when controls

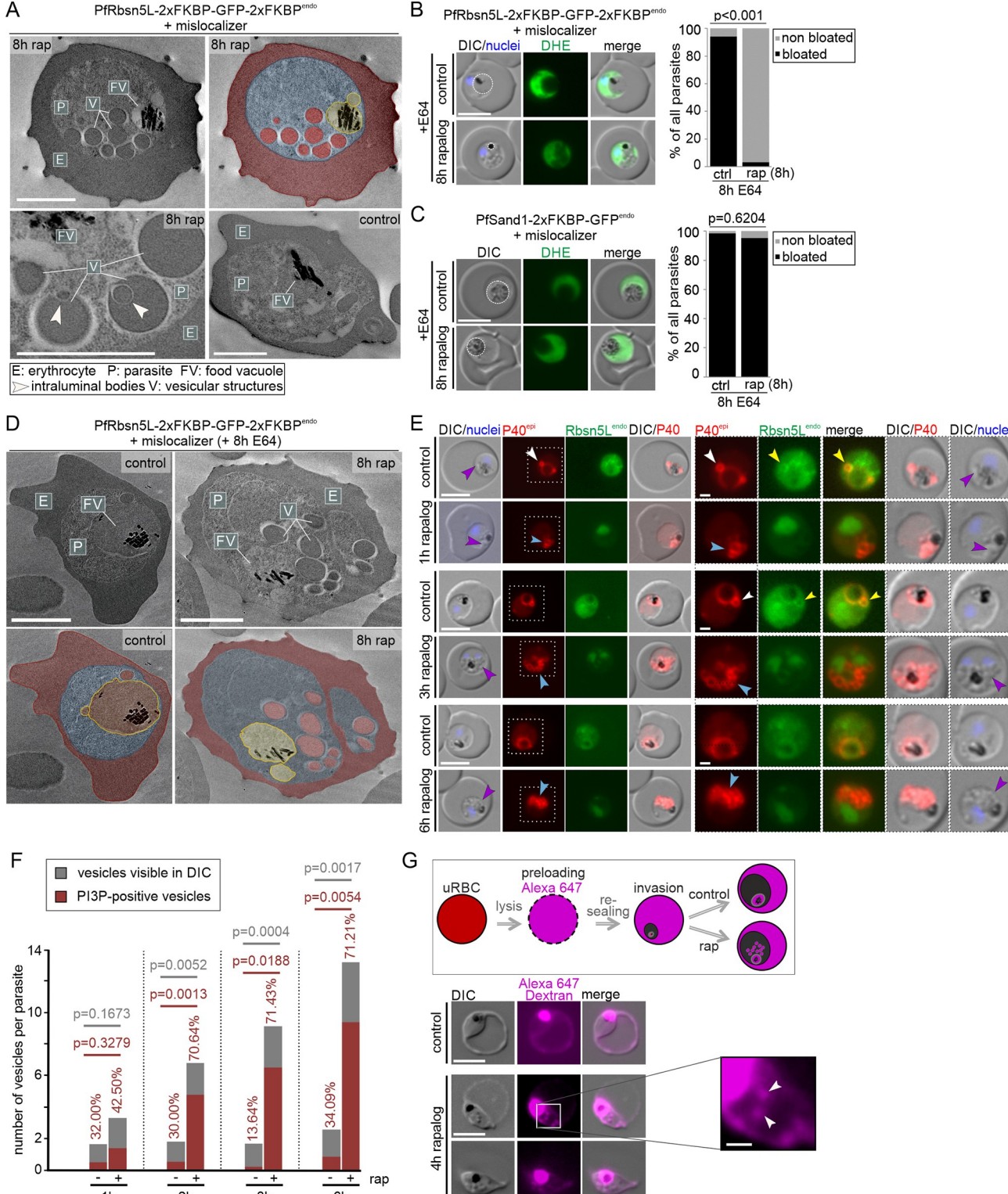

**Fig 2. Vesicular structures after PfRbsn5L-inactivation are host cell cytosol uptake intermediates. (A)** Electron microscopy images of PfRbsn5L knock-sideways (8 h rap) and control parasites. One representative image of $n$ = 42 (rapalog) and 32 (control) cells. Top right shows the top left image with false coloring. E, erythrocyte (red); P, parasite (blue); FV; food vacuole (yellow); V; vesicular structure (red). White arrows indicate putative intraluminal bodies. Scale bars, 2 μm. **(B, C)** Live-cell images of PfRbsn5L (B) and PfSand1 (C) knock-sideways (rap+) and controls (ctrl) in parasites treated with E64 (E64+). Left: Live-cell images of representative DHE-stained parasites. Right: Quantification of number of cells with bloated FVs.

Fisher's exact test. Pooled from $n = 3$ independent experiments with PfRbsn5L: 20, 26, and 22 cells (control) and 22, 20, and 20 cells (rapalog), and PfSand1endo: 22, 21, and 21 cells (control) and 29, 18, and 23 cells. *P*-values indicated. Rapalog was added in young trophozoites (see Materials and methods). **(D)** Electron microscopy images of PfRbsn5L-knock-sideways (8h rap) and control parasites treated 8 h with E64. False coloring and labels as in (B). Scale bars, 2 μm. **(E)** Live-cell images of PfRbsn5L knock-sideways (1 h, 2 h, 3 h, and 6 h rapalog) and control parasites, co-expressing the mScarlet tagged PI3P marker P40 (P40epi). White arrows: PI3P positive structures adjacent to the food vacuole. Yellow arrows: PfRbsn5L signal at circular structure adjacent to the food vacuole. Blue arrows: accumulations of PI3P near the food vacuole. Purple arrows: overlap of PI3P signal with the vesicle-like structures visible in the DIC. Merge, overlay of red and green channels. Areas in dashed boxes are magnified on the right. **(F)** Quantification of the number of vesicles in DIC (gray) and the number of PI3P positive vesicles (red) of cells imaged in (E). Pooled from $n = 2$ independent experiments with 15 (1 h), 11 (2 h), 13 (3 h), and 17 (6 h) (control) and 12 (1 h), 16 (2 h), 13 (3 h), and 15 (6 h) cells (rapalog), respectively. Two-tailed unpaired *t* test; *p*-values indicated. **(G)** Top: Schematic illustration of the experiment: RBCs are preloaded with fluorescent dextran (Alexa647); parasites invade and develop within preloaded RBCs cultivated in the presence or absence of rapalog (rap). Bottom: Live-cell images of preloaded infected RBCs with PfRbsn5L knock-sideways (4 h rapalog) and control parasites. White arrows: vesicular structures positive for Alexa647 dextran. Scale bar, 5 μm and 1 μm in the magnifications. Nuclei were stained with DAPI. DIC, differential interference contrast; endo, endogenous; epi, episomal; PfSand1endo, 2xFKBP-GFP tagged PfSand1 expressed from the endogenous locus [30]. The data underlying this figure can be found in S1 Data.

had progressed to schizonts and new rings, the parasites with inactivated PfRbsn5L were filled with vesicles, indicating that congestion of the cytoplasm with vesicles prevented successful completion of the cycle (S4C Fig).

Vesicles induced upon PfVPS45 inactivation were previously observed to be positive for PI3P, a characteristic feature of early endosome membranes [13]. To test if vesicles induced upon PfRbsn5L inactivation also share this endosomal feature, mScarlet tagged P40PX [3] was used as a marker for PI3P. In control parasites, PI3P was mainly found at the food vacuole membrane (Fig 2E) and sporadically at 1 or 2 small circular structures adjacent to the food vacuole (Fig 2E, white arrows), in agreement with previous reports [13,47]. We also noticed that PfRbsn5L accumulations appeared to be present at some of the PI3P-positive areas (Fig 2E, yellow arrows). We used confocal microscopy to better analyze this, which showed that 90.6% of the imaged parasites (48 of $n = 53$ cells) showed such an overlap, either directly at the PI3P delineated FV or at PI3P accumulations near the FV (S4D Fig). This indicated that PfRbsn5L is frequently present in some of the regions containing PI3P-positive membranes.

Upon inactivation of PfRbsn5L, increasing accumulations of PI3P near the food vacuole were seen over time (Fig 2E, blue arrows). The location of this signal overlapped with the vesicle-like structures visible by DIC (Fig 2E, purple arrows). Of the accumulating vesicles, 70.64% were PI3P positive 2 h after inducing inactivation of PfRbsn5L, a ratio that remained stable with increasing numbers of accumulating vesicles at 3 h and 6 h post induction (Fig 2F).

In order to show that these vesicular structures indeed contain host cell cytosol, we let the parasites invade and grow in erythrocytes preloaded with fluorescent dextran prior to the inactivation of PfRbsn5L (Fig 2G). The resulting vesicles were positive for fluorescent dextran, demonstrating that they are filled with host cell cytosol (Fig 2G, white arrow).

## Conditional inactivation of PfRab5b results in a phenotype resembling PfRbsn5L and PfVPS45 inactivation

VPS45 and Rbsn5 are typically found in a complex with Rab5 [33,48]. *P. falciparum* Rab5b possesses an N-terminal myristoylation site and lacks the usual C-terminal prenylation motif [49] (Fig 3A). This allowed us to modify the *rab5b* gene locus using the SLI system to generate a cell line (PfRab5bendo) endogenously expressing a C-terminally GFP-2xFKBP-tagged PfRab5b fusion protein (Figs 3B and S2A). In trophozoites, PfRab5b was localized at the food vacuole (Fig 3B, orange arrow) with occasional PfRab5b-positive circular signals adjacent to the food vacuole (Fig 3B, purple arrow). Furthermore, we found PfRab5b located at the plasma membrane (Fig 3B, white arrow) and the ER (Figs 3B, blue arrow, and S5A). In later stages, PfRab5b showed a pattern typical for the IMC (Figs 3B and S5B), potentially indicating an

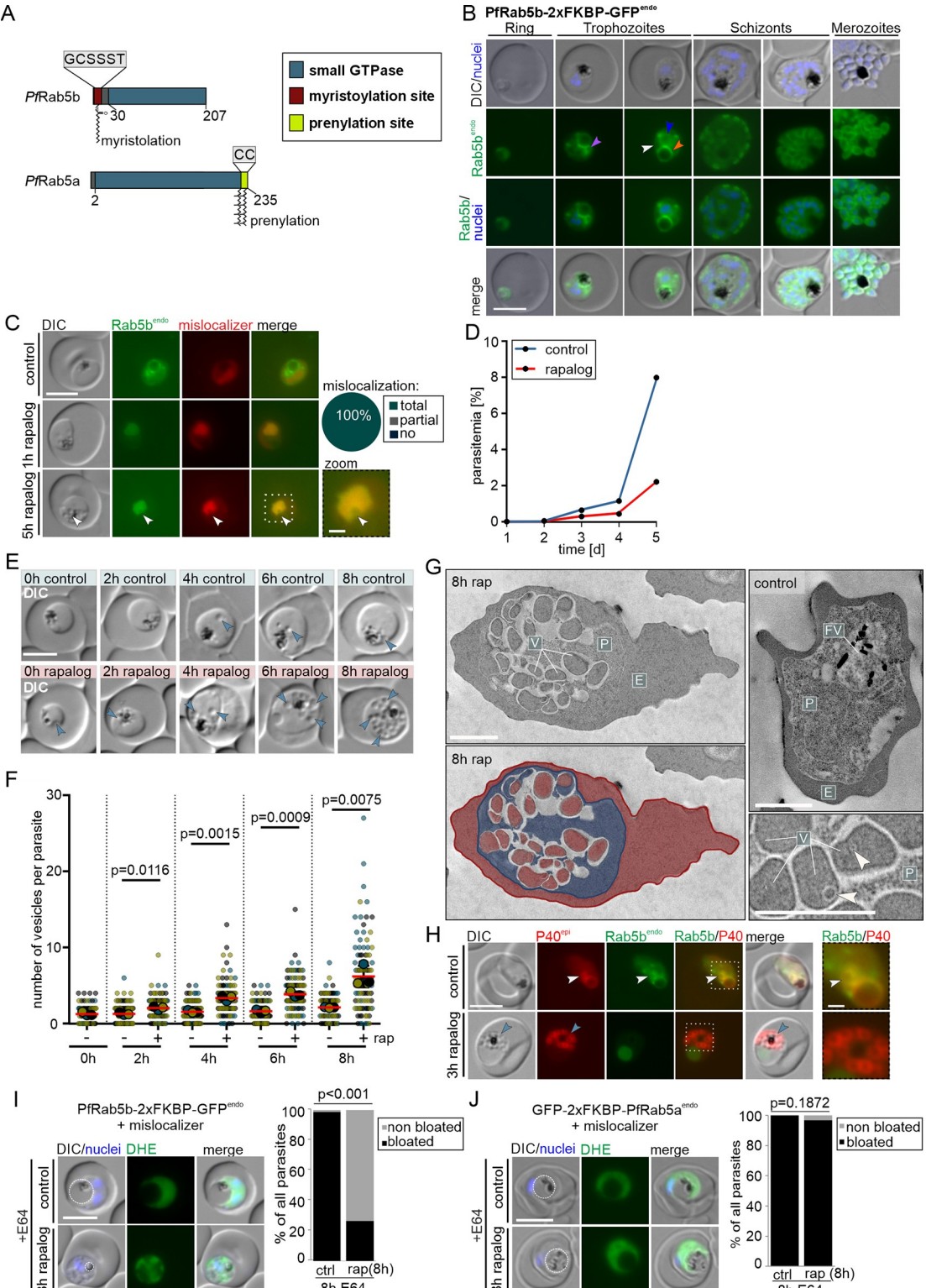

**Fig 3. PfRab5b is needed for host cell cytosol uptake. (A)** Domain architecture of *P. falciparum* Rab5b (PF3D7_1310600) and Rab5a. **(B)** Live-cell microscopy images of the indicated stages of PfRab5b^endo^ parasites. Orange arrow: signal at the food vacuole, purple arrow: circular signals adjacent to the food vacuole, white arrow: signal at the plasma membrane, blue arrow: ER localization. **(C)** Live-cell images of PfRab5b knock-sideways (1 h and 5 h rapalog) and control parasites. Knock-sideways was classified (pie chart) as complete (no signal detected outside the nucleus), partial (signal in the nucleus but also at original site), or

absent (no) mislocalization in $n$ = 55 parasites from 3 independent experiments. Zoom, enlargement of boxed region. **(D)** Flow cytometry-based growth curve over 2.5 growth cycles of PfRab5b knock-sideways (rapalog) compared to the control. One representative of $n$ = 3 independent experiments, all replicas shown in (S1G Fig). **(E)** Representative DIC images of live parasites 0 h, 2 h, 6 h, and 8 h after induction of knock-sideways of PfRab5b (rapalog) compared to control. Arrows, vesicular structures. **(F)** Quantification of number of vesicles per parasite in synchronous trophozoites 0 h, 2 h, 6 h, and 8 h after induction of PfRab5b knock-sideways. Data shown as superplots [77] from $n$ = 3 independent experiments (individual experiments are in blue ($n$ = 306 parasites), yellow ($n$ = 306 parasites), and black ($n$ = 306 parasites) (small dots), respectively; average of each experiment as large dot); two-tailed unpaired $t$ test of the means, $p$-values indicated; mean (red bar); error bars (black) show SD. **(G)** Electron microscopy images of PfRab5b knock-sideways (8 h rap) and control parasites. Bottom left shows the image from top left with false coloring. E, erythrocyte (red); P, parasite (blue); FV; food vacuole (yellow); V; vesicular structure (red). White arrows: putative intraluminal bodies. **(H)** Live-cell images of knock-sideways of PfRab5b (3 h rapalog) and control parasites, co-expressing mScarlet-tagged PI3P marker P40 (P40^epi). White arrows: PI3P positive structures adjacent to the FV. Blue arrows: accumulations of PI3P near the FV over time. The image on the right shows enlargement of the boxed region. **(I, J)** Live-cell images of PfRab5b (I) and PfRab5a (J) knock-sideways (8 h rapalog) treated 8 h with E64. Left: Live-cell images of DHE-stained parasites. Dashed circle highlights FV. Right: Quantification of the number of cells with bloated FVs. Fisher's exact test. Pooled from $n$ = 3 independent experiments (PfRab5b: 34, 34, and 34 cells (control) and 34, 34, and 34 cells (rapalog); PfRab5a: 23, 22, and 33 cells (control) and 22, 26, and 13 cells (rapalog)). $P$-values indicated. Scale bars in fluorescence microscopy images are 5 μm and 1 μm in enlargements and 2 μm in electron microscopy images. Nuclei in B, I, and J were stained with DAPI. DIC, differential interference contrast; endo, endogenous; epi, episomal; PfRab5b^endo C-terminally and Rab5a^endo N-terminally tagged with 2xFKBP-GFP expressed from endogenous locus. The data underlying this figure can be found in S1 Data.

additional function of PfRab5b in schizonts apart from HCCU, similar to what we previously reported for PfVPS45 [29].

To study the function of PfRab5b, we conditionally inactivated it by using knock-sideways with a nuclear mislocalizer. After the addition of rapalog, PfRab5b was efficiently mislocalized to the nucleus within 1 h (Fig 3C). To determine the importance of PfRab5b in parasite blood stages, we tracked the parasitemia of these parasites grown in presence and absence (control) of rapalog over 5 days by flow cytometry. Inactivation of PfRab5b resulted in a growth defect when compared to control parasites (Figs 3D and S2D), indicating that PfRab5b function is important for asexual blood stage growth.

Similar to what we observed for PfRbsn5L (Fig 1F, blue arrows), and previously for PfVPS45 [13], the inactivation of PfRab5b led to an accumulation of vesicular structures in the parasite cell (Fig 3E). Crescent-shaped signals of the mislocalizer and the PfRab5b at some of the nuclei indicated vesicles close to the nucleus (Fig 3C, white arrows). The number of vesicles per parasite increased over time with an average of 6.53 ± 1.28 after 8 h of PfRab5b inactivation (Fig 3F), less than after the inactivation of PfRbsn5L (Fig 1G) or PfVPS45 [13]. The number of induced vesicles also varied more between the individual parasites than they did upon PfRbsn5L and PfVPS45 inactivation: while some parasites showed the formation of many vesicles, other parasites contained none or only a few vesicles (Fig 3F). Some differences were also seen in the stage-specific growth phenotype compared to PfRbsn5L inactivation (S3 Fig), as there was no loss of ring stage viability but a similar (although less profound) effect to that seen with PfRbsn5L inactivation during trophozoite to schizont development (S5C Fig) that led to aberrant parasites filled with vesicles (S5D Fig).

Electron microscopic examinations of parasites after PfRab5b inactivation showed that the vesicular structures present within the parasites contained material that appeared to correspond to host cell cytosol (Fig 3G) with some harboring smaller internal vesicles (Fig 3G, white arrows), similar to the PfRbsn5L and PfVPS45 inactivation phenotype. Episomal expression of P40PX showed in the control parasites that the PI3P-positive circular structures occasionally seen adjacent to the food vacuole colocalized with those observed for PfRab5b (Fig 3H, white arrows and Fig 3B, purple arrow). After inactivation of PfRab5b, increasing accumulations of PI3P signals near the food vacuole were observed, overlapping with the vesicle-like structures visible in the DIC images (Fig 3H, blue arrows).

Next, we directly assessed the role of PfRab5b in HCCU by using bloated food vacuole assays. Inactivation of PfRab5b resulted in non-bloated food vacuoles in approximately 73% of the parasites, demonstrating the involvement of PfRab5b in HCCU (Fig 3I). In contrast, inactivation of PfRab5a showed no significant defect on food vacuole-bloating (Fig 3J). Previous work had indicated that PfRab5a has no role in HCCU but this was not directly tested [30]. Taken together, our bloated food vacuole assays indicate that PfRab5b but not PfRab5a is needed for HCCU.

## PfRbsn5L, PfRab5b, and PfVPS45 inactivation leads to HCC-filled vesicles with no connection to the host cell

To evaluate whether the observed vesicular structures upon PfRbsn5L and PfRab5b inactivation are indeed vesicles and are not still connected to the host cell cytosol—e.g., are cytostomes—parasites were treated with saponin to remove the host cell cytosol content followed by an anti-hemoglobin IFA to detect the vesicular content (Fig 4A). As a positive control, we inactivated PfVPS45 [13]. We observed an average number of 9.44 ± 0.08, 8.23 ± 0,54, and 4.48 ± 0.47 of spherical-shaped anti-hemoglobin-positive individual areas after 6 h of PfVPS45, PfRbsn5L, and PfRab5b inactivation, respectively (Fig 4B, white arrows and 4C). In contrast, the controls (the matched parasites where knock-sideways was not induced) showed significantly fewer (0.61 ± 0.14, 0.26 ± 0.04, and 0.33 ± 0.06) individual anti-hemoglobin-positive areas (Fig 4B and 4C). However, the control parasites showed a notably larger hemoglobin-positive area corresponding to the DV (Fig 4B, position of food vacuole indicated by blue arrows). These findings demonstrate that the structures appearing after the inactivation of these proteins contain HCC, but are not connected to the host cell, and consequently, they can be designated as vesicles.

These experiments were confirmed by electron microscopy with saponin-lysed parasites after either PfRbsn5L or PfRab5b inactivation which revealed that the vesicles induced in the parasite were filled with electron-dense material while the host cell was translucent, confirming the successful release of the HCC from the host cell while the vesicles remained filled (Fig 4D). In addition, it is evident in these parasites that the vesicles are enclosed by a double membrane (Fig 4D, yellow and blue arrows), as described before for vesicles upon PfVPS45 inactivation and congruent with a cytostomal origin which is an invagination of both the PPM and PVM.

## PfRab5b and PfVPS45 are interaction partners of PfRbsn5L

In model systems Rbsn5, VPS45, and Rab5b function in one complex. To analyze whether these proteins could function in a similar manner in malaria parasites, we initially conducted localization studies. Episomal co-expression of PfVPS45-mCh[epi] in PfRbsn5L-2xFKBP-GFP-2xFKBP[endo] parasites showed overlapping signals of both proteins (Figs 5A, white arrows and S6A) and co-expression of PfRab5b-mCh[epi] in PfRbsn5L-2xFKBP-GFP-2xFKBP[endo] (Figs 5B, white arrows and S6B) and in PfVPS45-2xFKBP-GFP[endo] (Figs 5C, white arrows and S6C) parasites indicated some localization of PfRbsn5L and PfVPS45 to PfRab5b positive membranes. To corroborate the co-localization of PfRbsn5L with Rab5b, we carried out confocal microscopy which confirmed overlapping signals of both proteins in proximity of the food vacuole (S6D–S6F Fig).

To determine if PfRbsn5L interacts with PfVPS45 and PfRab5b, we performed co-immuno-precipitations (CoIPs) (Figs 5D, 5E, S7, and S8). Immunoprecipitation of the endogenously GFP-tagged PfRbsn5L co-purified PfVPS45-mCh[epi] (Figs 5D and S7) and PfRab5b-mCh[epi] (Figs 5E and S8) in the respective cell lines while the control protein BIP was not enriched. These data indicate that PfRbsn5L interacts with PfRab5b and PfVPS45 in *P. falciparum*

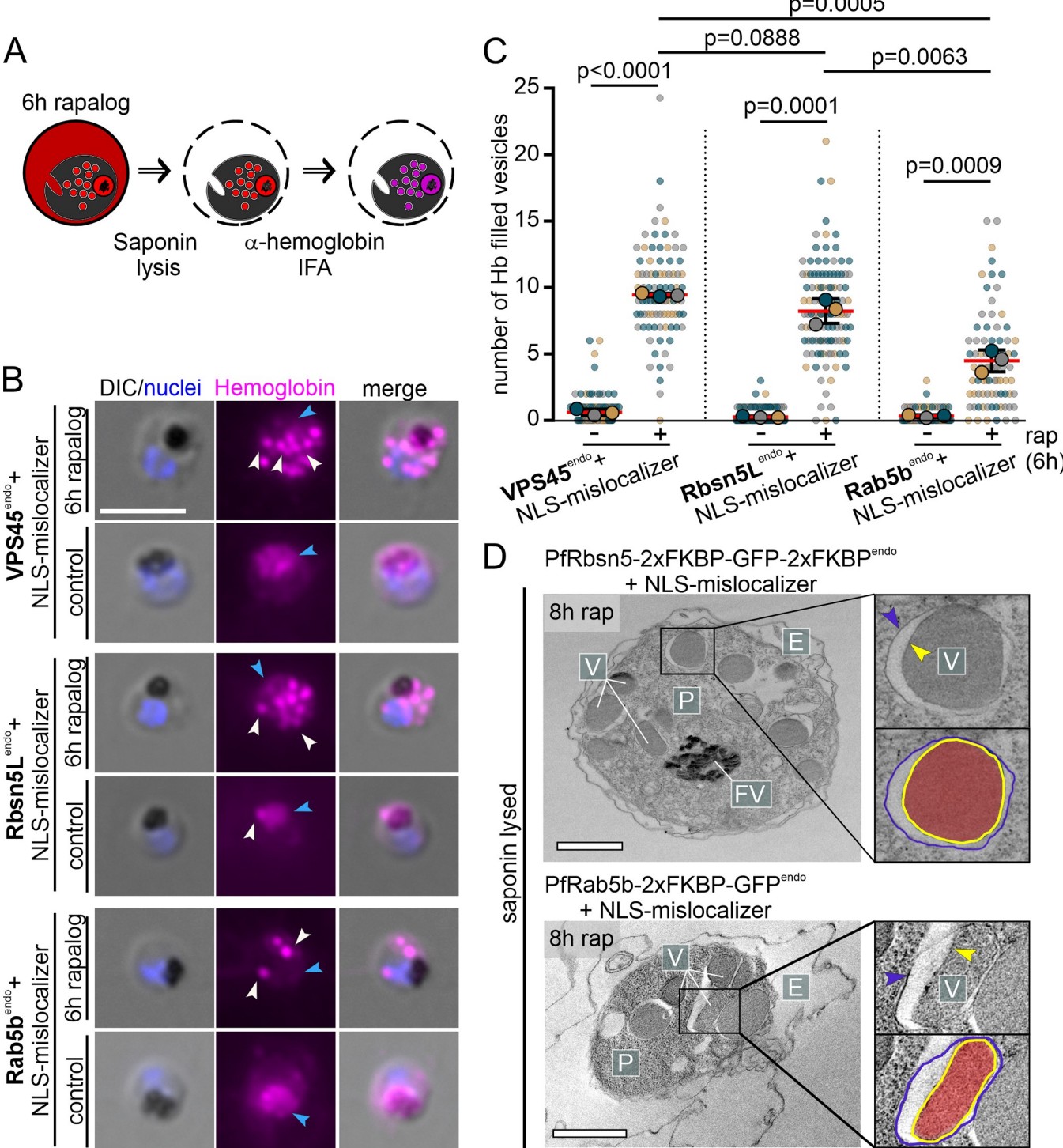

**Fig 4. PfRbsn5L, PfVPS45, and PfRab5b inactivation-induced vesicles are hemoglobin-filled and not connected to the host cell. (A)** Schematic illustration of experimental procedure for IFA after saponin treatment. **(B)** Microscopy images of α-hemoglobin (Hemoglobin) IFAs of formaldehyde-fixed, PfVPS45endo, PfRbsn5Lendo, and PfRab5bendo knock-sideways (6 h rapalog) and control parasites (cell lines indicated). White arrows: accumulated hemoglobin positive signals, blue arrows: DV. Nuclei were stained with DAPI. DIC, differential interference contrast. Scale bar, 5 μm. **(C)** Quantification of hemoglobin positive foci per cell 6 h after induction of the knock-sideways (rap) from B. Superplot from *n* = 3 independent experiments with PfVPS45: 32, 31, and 39 cells (control) and 24, 29, and 34 cells (rapalog), PfRbsn5L: 27, 60, and 49 cells (control) and 23, 44, and 47 cells (rapalog) and PfRab5b: 24, 24, and 20 cells (control) and 26, 32, and 21 cells (rapalog), respectively; small dots show individual cells, large dots show the mean, color-coded by experiment; two-tailed unpaired *t* test of the means, *p*-values indicated; mean, red bar; error bars, black (SD). **(D)** Electron microscopy images of saponin-treated PfRbsn5L and PfRab5b knock-sideway

parasites (8 h rap). Representatives of $n$ = 12 (PfRbsn5L) and 50 cells (PfRab5b). E, erythrocyte; P, parasite; FV; food vacuole; V; vesicular structure. Scale bars, 2 μm. The data underlying this figure can be found in S1 Data.

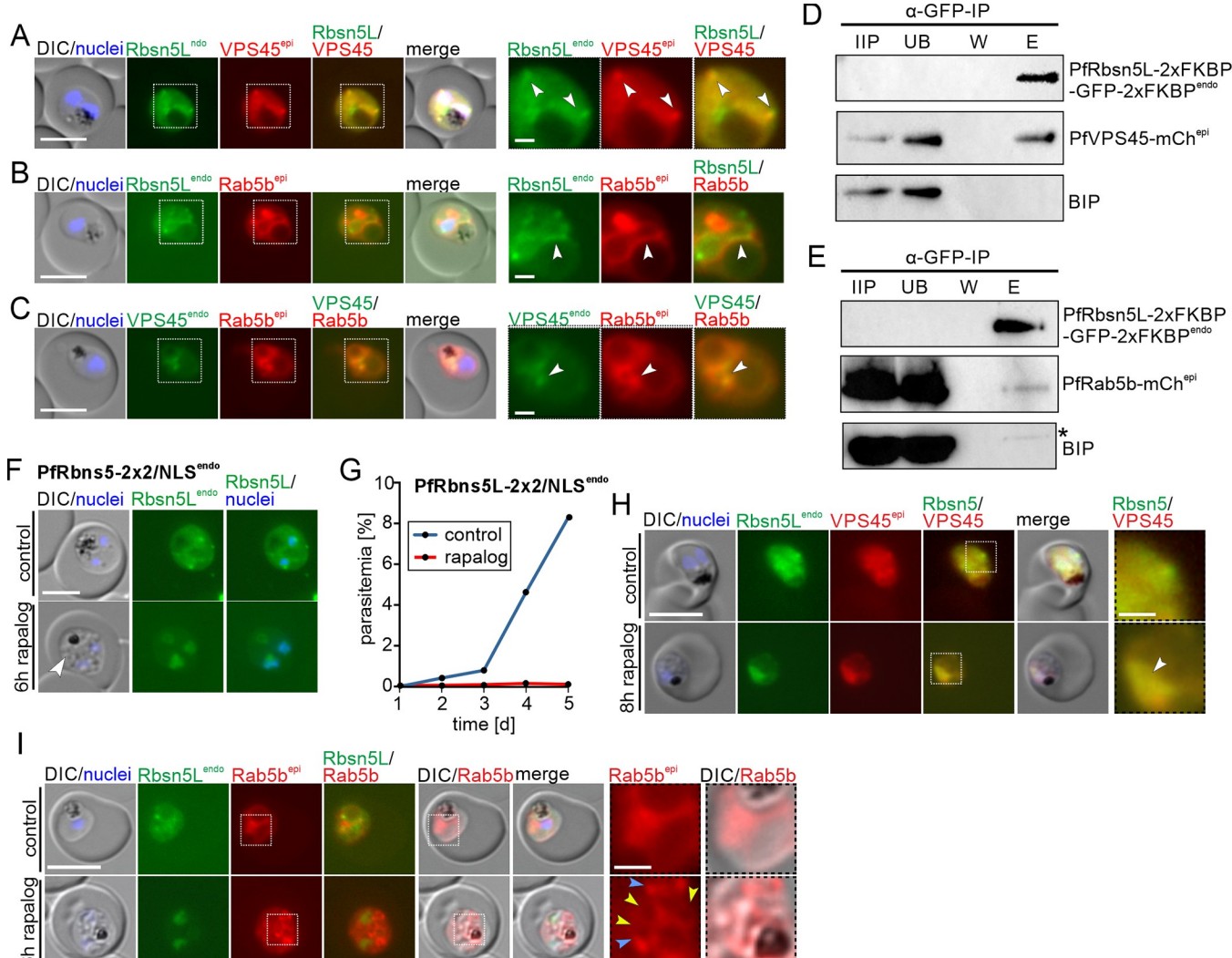

**Fig 5. PfRab5b and PfVPS45 interact with PfRbsn5L. (A–C)** Live-cell microscopy images of trophozoites of PfRbsn5L[endo] parasites co-expressing PfVPS45[epi] (A), PfRab5b[epi] (B) or VPS45[endo] parasites, co-expressing PfRab5b[epi] (C). White arrows: overlapping of PfRbsn5L[endo] or PfVPS45[endo] and PfRab5b[epi] signals. Other stages of A–C shown in S3 Fig. **(D, E)** Immunoprecipitation (IP) of PfRbsn5L[endo] parasites co-expressing PfVPS45[epi] (D) or PfRab5b[epi] (E). IIP, IP-input extract; UB, unbound (total extract after IP); W, last wash; E, eluate. Asterisk on BIP blot shows degradation band of GFP-fusion protein from previous exposure (replicas and full blots in S4 and S5 Figs and S1 Raw Images). One representative of $n$ = 3 independent experiments (all replicas and complete blots shown in S4 and S5 Figs). **(F, G)** Validation of system for knock-sideways of a POI (PfRbsn5L) with a single integrated plasmid (PfRbsn5L-2xFKBP-GFP-2xFKBP[endo]_*nmd3*'NLS-FRB-T2A-hDHFR, named PfRbsn5L-2x2/NLS[endo] parasites). Live-cell images (F) and flow cytometry-based growth curve over 2.5 growth cycles (G) of induced knock-sideways of PfRbsn5L (rapalog) and control in these parasites. Growth experiment shows 1 representative of $n$ = 3 independent experiments, all replicas shown in S1H Fig. Arrow, induced vesicles. **(H)** Live-cell microscopy images of PfRbsn5L knock-sideways (8 h rapalog) and controls using the PfRbsn5L-2x2/NLS[endo] parasites co-expressing PfVPS45[epi]. Representative images of $n$ = 10 (control) and $n$ = 15 (rapalog) cells. White arrow: Localization of PfVPS45 and PfRbsn5L to the nucleus upon PfRbsn5L inactivation. **(I)** Live-cell microscopy images of PfRbsn5L knock-sideway (6 h rapalog) and controls of PfRbsn5L-2x2/NLS[endo] parasites co-expressing PfRab5b[epi]. Yellow arrows: PfRab5b[epi] signal surrounding vesicles observed by DIC. Blue arrows: PfRab5b accumulations. Representative images of $n$ = 2 independent experiments with 11, 5 (control) and 19, 11 (6 h rapalog) cells. Dashed boxes in the images in A–C, H, and I are shown as magnification on the right. DIC, differential interference contrast; endo, endogenous; epi, episomal. The scale bars, 5 μm and 1 μm in the magnifications. Nuclei were stained with DAPI. The data underlying this figure can be found in S1 Data.

parasites and therefore support the hypothesis of their function in a conserved complex in malaria parasites. This interaction further strengthens the idea that the protein herein assigned as PfRbsn5L indeed is the Rbsn5 equivalent of the parasite. We note that proportionally less Rab5b was co-immunoprecipitated compared to VPS45 which could indicate a more transient interaction or a weaker binding (compare Fig 5D with 5E, and S7 Fig with S8).

Next, we assessed the fate of PfVPS45 and PfRab5b after inactivation of PfRbsn5L. However, as there are only a limited number of resistance markers available, we first had to devise a system for knock-sideways of a POI (in this case PfRbsn5L) with a single integrated plasmid using SLI, resulting in cell line PfRbsn5L-2x2/NLS$^{endo}$ (Figs 5F, S2A, and S9A). Testing the effectivity of this approach showed that upon addition of rapalog, PfRbsn5L was successfully mislocalized to the nucleus (Fig 5F), vesicular structures accumulated as evident in the DIC images (Fig 5F, white arrow), and monitoring parasitemia after conditional inactivation over 5 days showed a drastic growth defect in comparison to the control parasites (Figs 5G and S2D). Hence, results comparable to those using an episomally overexpressed mislocalizer (Fig 1D and 1E) were obtained with this single plasmid system with which the mislocalizer is expressed from a single copy in the genome.

Next, we assessed the effect of PfRbsn5L inactivation on episomally expressed mCherry tagged PfVPS45 and PfRab5b. Mislocalization of PfRbsn5L into the nucleus resulted in a co-mislocalization of PfVPS45 (Fig 5H), indicating that PfVPS45 interacts with PfRbsn5L and is indirectly pulled into the nucleus. In contrast, PfRab5b was not co-mislocalized upon PfRbsn5L inactivation but led to accumulations of PfRab5b signal in the parasite (Fig 5I, blue arrows) that were absent in the control (Fig 5I). These accumulations were at the emerging vesicles (evident in the DIC images) and also appeared to be surrounding the vesicles with a faint signal (Fig 5I, yellow arrows), similar to what was observed for PI3P (Fig 2E and [13]). These results suggest localization of PfRab5b on the induced HCC-filled vesicles, further underlining the endosomal character of these vesicles and the role of PfRab5b in endosomal transport in malaria parasites. Overall, these data indicate that PfRbsn5L, PfVPS45, and PfRab5b function together in endosomal transport in malaria parasites and that PfRbsn5L and PfVPS45 form a more stable complex than they do with their interaction partner PfRab5b.

## Inactivation of the cytostome protein KIC7 prevents generation of PfRbsn5L- and cytochalasin D-induced vesicles

Currently, there are 2 categories of proteins affecting HCCU, those that act early when endocytic containers are generated and do not lead to HCC-filled vesicles when inactivated [3,15] and those that presumably act later and therefore lead to HCC-filled intermediates when inactivated ([13]; this work). However, if these are indeed serial steps in the same pathway has not been tested. Here, we established a system to conditionally inactivate 2 different proteins at the same time (S9B Fig) to test this by simultaneously inactivating KIC7 (a protein at the Kelch13 compartment that is needed for the early part of HCCU at the PPM [3]) and PfRbsn5L (needed for transport of HCC to the food vacuole) (Figs 6B and S2A). In case these pathways are connected, inactivation of KIC7 should also reduce the number of vesicles resulting from PfRbsn5L-inactivation, as less material enters the pathway that subsequently depends on PfRbsn5L (S9C Fig). It should be noted that this system is not orthogonal and hence both proteins are inactivated around the same time. To control for independent effects, we again used the non-endocytosis, essential vesicle trafficking protein PfSand1 (Figs 6C and S2A).

In all cell lines inactivation of the target proteins occurred as expected (Fig 6A–6C). Inactivation of PfRbsn5L alone (Fig 6A) resulted in 8.81 ± 1.08 vesicles per parasite, whereas simultaneous inactivation of KIC7 and PfRbsn5L led to a significantly reduced accumulation of

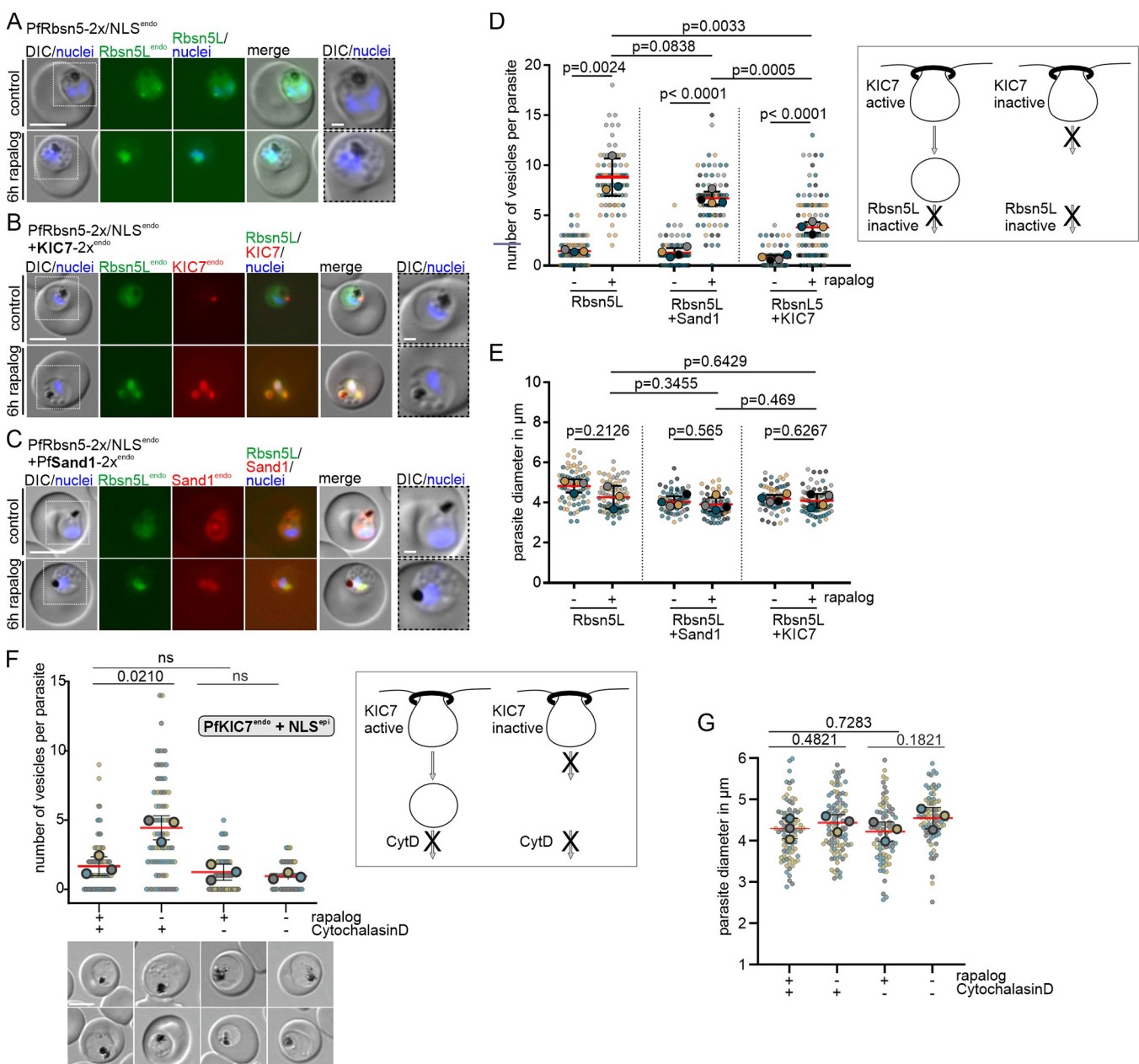

**Fig 6. Inactivation of PfRbsn5L or cytochalasin D treatment together with KIC7 inactivation leads to reduced vesicle accumulation. (A–C)** Knock-sideways (6 h rapalog) of highly synchronous parasites (30–33 h post invasion) of PfRbsn5L-2x/NLS[endo] (using cell line PfRbsn5L-2xFKBP-GFP/NLS-FRB[endo]), PfRbsn5L-2x/NLS[endo]+ KIC7-2x[endo] (using cell line PfRbsn5L-2xFKBP-GFP/NLS-FRB[endo]+ KIC7-2xFKBP-mCh[endo]), and PfRbsn5L-2x/NLS[endo]+ PfSand1-2x[endo] (using cell line PfRbsn5L-2xFKBP-GFP/NLS-FRB[endo]+ PfSand1-2xFKBP-mCh[endo]) and control (without rapalog) parasites. Nuclei were stained with DAPI. DIC, differential interference contrast. Dashed boxes in the images are areas shown as magnification on the right. Scale bar, 5 μm. **(D, E)** Superplots showing quantification of the vesicle number per parasite (D) and parasite diameter (E) of the cells shown in A–C. Data from $n = 3$ (PfRbsn5L-2x/NLS [endo]) or $n = 4$ (PfRbsn5L-2x/NLS[endo]+ KIC7-2x[endo] and PfRbsn5L-2x/NLS[endo] + PfSand1-2x[endo]) independent experiments. PfRbsn5L-2x/NLS[endo]: 21 (gray dots), 31 (yellow dots) and 30 (blue dots) cells (control) and 23, 30, and 24 cells (rapalog), PfRbsn5L-2x/NLS [endo] + KIC7-2x[endo]: 15 (black dots), 19 (gray dots), 18 (yellow dots), and 19 (blue dots) cells (control) and 21, 20, 22, and 35 cells (rapalog), and PfRbsn5L-2x/NLS [endo] + PfSand1-2x[endo]: 14 (black dots), 16 (gray dots), 13 (yellow dots), and 9 (blue dots) cells (control) and 20, 12, 9, and 26 cells (rapalog), respectively. Small dots represent 1 parasite, large dots the mean of each independent experiment; two-tailed unpaired $t$ test of the means, $p$-values indicated; mean (red); error bars (black) show SD. Scheme besides D shows expected outcome if cytostome (top) function generates endosomal cargo needing Rbsn5L for transport (bottom). **(F, G)** Superplots showing quantification of the number of vesicles in PfKIC7endo + NLS parasites (F) and diameter of these parasites (G). Data from $n = 3$ independent experiments. Control: 22 (gray dots), 32 (yellow dots), and 27 (blue dots) cells; rapalog: 26, 32, and 27 cells; cytochalasin D: 26, 32, and 27 cells; rapalog and cytochalasin D: 26, 29, and 27 cells. Small dots represent 1 parasite, large dots the mean of each independent experiment; two-tailed unpaired $t$ test of the means, $p$-values indicated; mean (red); error bars (black) show SD. Scheme besides F as in D. The data underlying this figure can be found in S1 Data.

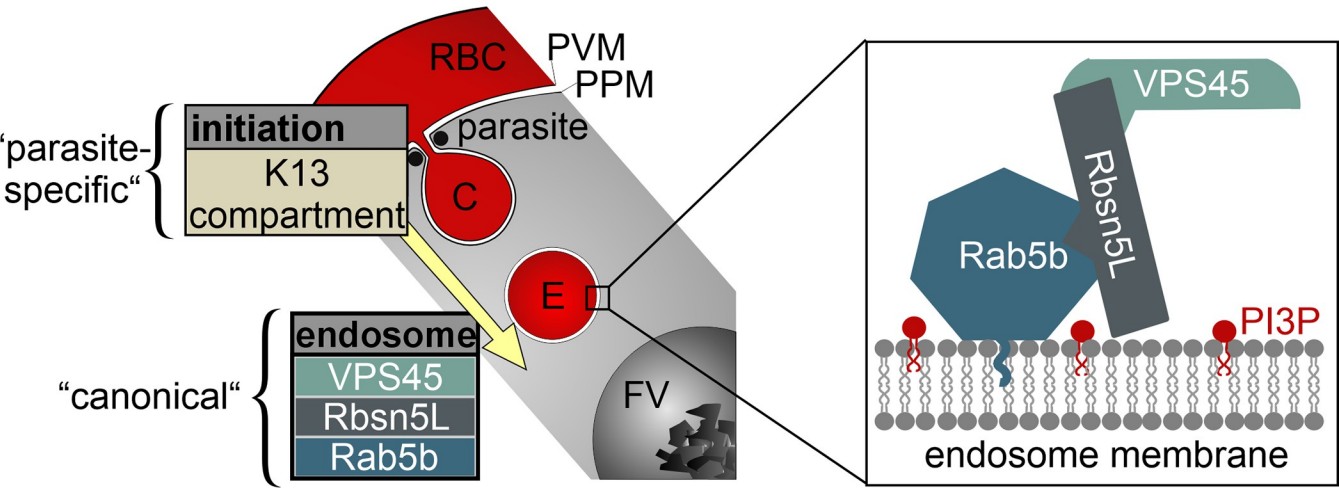

**Fig 7. Schematic of proteins in HCCU.** Section of infected RBC showing "initiation" phase of endocytosis (with strongly parasite-specific characteristics [3,15]) feeding into an endosomal transport pathway that depends on a functional complex of VPS45, Rbsn5L, and Rab5b (enlargement box) which indicates conserved elements playing a role in this phase of HCCU. FV, food vacuole; C, cytostome; E, endosome; PI3P, phosphatidylinositol-3-phosphate; RBC, red blood cell; PVM, parasitophorous vacuolar membrane; PPM (parasite plasma membrane); HCCU, host cell cytosol uptake; yellow arrow, direction of the pathway.

3.79 ± 0.27 vesicles per parasite (Fig 6B and 6D). This was also significantly less than when PfRbsn5L and PfSand1 were inactivated which resulted in 6.69 ± 0.33 vesicles per parasite (Fig 6C and 6D). The diameter of the parasites did not differ significantly to the controls in these experiments (Fig 6E), excluding reduced viability as a cause for the differences in the number of vesicles between the different parasite lines.

In order to corroborate these findings, we used the chemical inhibitor cytochalasin D instead of PfRbsn5L inactivation. Cytochalasin D depolymerizes actin and results in the accumulation of hemoglobin filled vesicles in the parasite, indicating it inhibits endosomal transport [12,22]. While in the control, cytochalasin D treatment resulted in the accumulation of vesicles in the parasite, conditional inactivation of KIC7 prevented the occurrence of the cytochalasin D-induced vesicles (Fig 6F). This indicated that the vesicles arising from cytochalasin D treatment arise from functions at the cytostome. Again, this was unlikely due to a viability loss over the assay time as there was no significant difference between the parasite size of the different conditions (Fig 6G).

These results indicate that KIC7 and its function at the cytostome are necessary to generate the vesicles that accumulate upon PfRbsn5L inactivation or cytochalasin D treatment. Overall, these experiments provide evidence that the cytostome lies upstream of a pathway with endosomal characteristics that transports HCC to the food vacuole (Fig 7).

## Discussion

The first protein experimentally shown to be needed for endosomal transport in malaria parasites was the parasite's orthologue of VPS45 [13]. In model organisms, VPS45 functions together with Rbsn5 and Rab5 in fusion events necessary for endosomal transport processes [33,34,48,50,51]. However, no Rbsn5 is evident in the parasite's annotated genome and the only Rab5 for which functional data are available (PfRab5a) did not indicate a role in endocytosis [30]. Hence, it was unclear to what extent the function of PfVPS45 was indicative of a conservation of the endosomal vesical trafficking system in malaria parasites, particularly as the initial phase of endocytosis at the PPM shows evidence of strong parasite-specific

adaptations [3,15]. Here, we identified a Rbsn5-like protein in *P. falciparum* and show that it functions with PfVPS45 in endosomal transport. We also show that PfRab5b but not PfRab5a is involved in this process, overall indicating that a PfVPS45-PfRbsn5L-PfRab5b complex operates in endosomal transport of hemoglobin to the food vacuole in malaria parasites. While PfRbsn5L displays only limited sequence identity with Rbsn5 of model organisms, its experimentally determined function and interactors nevertheless indicate that endosomal maturation and transport of the parasite contains elements resembling those of model organisms. This similarity is strengthened by the finding that the hemoglobin-filled vesicles accumulating when PfRbsn5L, PfRab5b, or PfVPS45 are inactivated are positive for PI3P and some contain internal vesicles resembling intraluminal bodies ([13]; this work). In addition, we here show that PfRab5b becomes enriched in the area surrounding the hemoglobin-filled vesicles induced upon PfRbsn5L inactivation and is located at PI3P positive membranes close to the food vacuole at steady state, further supporting their endosomal character and the role of PfRab5b in endosomal maturation. Overall, our findings indicate that despite the atypical N-terminal instead of C-terminal modification [49], PfRab5b has a role akin to that of Rab5 isoforms in other organisms [52–54]. However, there are also important differences. We did not find any evidence for PI3P binding of the PfRbsn5L FYVE domain which contains key differences in the amino acids known to be needed for PI3P binding in other FYVE domains. Interestingly, the same amino acids were also not conserved in the FYVE domain of human protrudin which was found to preferentially bind other phosphoinositides than PI3P [41] and we here found no experimental evidence for a recruitment of the PfRbsn5L FYVE domain to PI3P containing areas in the cell. As PfRbsn5L localized only with some of the PI3P positive areas in the parasite, it is possible that the FYVE domain alone does not suffice for PI3P binding, permitting a more nuanced, function-dependent targeting. In contrast, FCP, which may correspond to the parasite's EEA1, is located around the food vacuole [36], does contain a fully canonical FYVE domain (S1A Fig). It is therefore possible that these differences underlie a functional separation of the 2 FYVE domain proteins of the parasite.

It is also possible that the PfRbsn5L FYVE domain does not bind phosphoinositides at all or that it binds a different phosphoinositide than PI3P. If the latter is the case, it likely is not PI4,5P2, PI4P, or PI5P which were detected at defined sites such as the plasma membrane and the Golgi of the parasite [55]. The FYVE domain of protrudin was shown to bind PI4,5P2, PI3,4P2, and PI3,4,5P3 [41]. PI3,4P2 and PI3,4,5P3 were detected in *P. falciparum*-infected RBCs [47], but no accumulation at a cellular structure was observed when specific sensors for these phosphoinositides were expressed in the parasite [55]. Given the apparent difficulties in detecting these phosphoinositide species with overexpressed sensors in the parasite, it remains possible that the FYVE domain of PfRbsn5L binds one of these species even though the overexpressed double PfRbsn5L FYVE domain appeared to be cytoplasmic.

We also noted a nuclear localization of PfRbsn5L. Previous work has shown a propensity for the protrudin FYVE domain to locate to the nucleus and there is also a FYVE domain protein with a function at the centrosome [41,56]. However, at present, the functional relevance for nuclear localization of PfRbsn5L is unclear. It is possible that PfRbsn5L has additional functions than in endosomal transport but the observed phenotype of the PfRbsn5L knocksideways indicates that its endosomal function is a cause of the observed growth phenotype.

In the apicomplexan parasite *T. gondii* it has been proposed that endosomal transport and secretion intersect at the endosomal-like compartment (ELC) located close to the Golgi [57]. Similarly to PfVPS45 [13], some PfRbsn5L was located proximal to the Golgi marker GRASP. While no ELC has so far been defined in *P. falciparum* parasites, one possibility is that the Golgi-proximal location of PfVPS45 and PfRbsn5L corresponds to an equivalent site. In yeast and other model organisms, VPS45 is believed to be required for Golgi-to-vacuole transport

and, when inactivated, results in a similar phenotype to that observed in *P. falciparum* parasites [13,19,20,58]. The *trans*-Golgi proximal location may therefore indicate that PfVPS45 and PfRbsn5L are needed for a Golgi to endosome vesicular exchange route that adds lytic enzymes and recycles components not destined for the food vacuole and that absence of this route arrests endosomal maturation. This hypothesis is congruent with the lack of detectable digested hemoglobin in the accumulated vesicles. However, only about 50% of the PfRbsn5L foci were close to GRASP, and PfVPS45 and PfRbsn5L are also found at accumulations at or near the food vacuole. This would speak more for a function where they are directly involved in fusion events that deliver endosomal HCC-filled vesicles to the food vacuole. In further support of this, we here also found that the food vacuole proximal accumulations of PfRbsn5L frequently overlapped with some of the PI3P positive areas, indicating that they contain endosomal structures. It is puzzling why the food vacuole of malaria parasites is the major site containing PI3P, as in other organisms PI3P typically is an early endosomal marker. The lack of PfRbsn5L's FYVE domain to bind PI3P may permit the PfVPS45-PfRbsn5L-PfRab5b fusion complex to act in a transport process upstream of the food vacuole, leading to fusion with the food vacuole rather than being fully recruited to the food vacuole membrane itself. Future work will be needed to better understand the general vesicle pathways in the parasite to provide a framework to understand endosomal transport and secretion in malaria parasites.

In contrast to the findings with the proteins studied in this work, the initial phase of hemoglobin endocytosis at the PPM seems to show strong differences to that in model organisms [3,15]. A seemingly permanent structure termed the cytostome, an invagination of the PPM (in *Plasmodium* parasites also the PVM) containing a number of proteins in a ring structure around the neck of the invagination (the Kelch13 compartment) was implicated in this step in malaria and *T. gondii* parasites [3,4,8,16–18]. It has so far not been experimentally shown that the step involving Kelch13 compartment proteins is connected to the pathway that is affected when PfVPS45, PfRbsn5L, or PfRab5b is inactivated, although both processes prevent hemoglobin from reaching the food vacuole [3,13]. Taking advantage of one of the proteins needed for hemoglobin endocytosis at the cytostome, the Kelch13 compartment protein KIC7, together with PfRbsn5L-inactivation, we here provide first evidence for a link between these 2 steps. There are some limitations of the system we used to study this, as both PfRbsn5L and KIC7 are inactivated at the same time. We therefore also used cytochalasin D which also causes the accumulation of hemoglobin-filled vesicles [12,22]. These data support the model that the hemoglobin-filled vesicles that occur after PfVPS45, PfRbsn5L, or PfRab5b inactivation derive from internalization events depending on the K13 compartment proteins at the PPM, providing evidence that cytostome function connects into endosomal transport. The results with cytochalasin D also further support a role of actin in endosomal transport, congruent with recent work implicating a Myosin in this process [15].

A large body of data suggests that many proteins that in other organisms function in the endosomal system were repurposed for secretory functions in apicomplexan parasites [8,25–29,59–63]. One example of such a protein is PfRab5a, for which we here directly show that it has no function in endocytosis. Also, PfSand1, in other systems a GEF of the late endosome marker PfRab7 [64,65], did not influence endocytosis in trophozoites when inactivated. In contrast, PfVPS45, PfRbsn5L, and PfRab5b are needed for endocytosis. These proteins, therefore, do not appear to have been repurposed. However, we previously showed that PfVPS45 also has a role in schizont-stage parasites where its inactivation impacted the biogenesis of the IMC and the formation of invasive merozoites, suggesting a role in secretory processes during cytokinesis [29]. It is therefore possible that proteins of the parasite were not only repurposed but that there also are stage-specific differences in function with individual proteins involved in endocytosis and secretory processes in different life cycle stages. This might also explain

why we found PfRab5b at the IMC in schizonts, despite its involvement in endocytosis in trophozoites. In light of this, the rapid inactivation kinetics of knock-sideways achieved with the proteins studied here are an advantage, as this permits to pinpoint stage-specific phenotypes and poses less danger of indirect effects due to imbalances arising from disturbances in other parts of the interconnected cellular vesicular trafficking pathways.

Overall, this work indicates that at least part of the endosomal system of malaria parasites contains elements that resemble that of model organisms. The identification of proteins in endosomal transport is important due to the critical role of HCCU for malaria blood stage parasites and the importance of its endpoint—hemoglobin digestion in the food vacuole—for action of and resistance to antimalarial drugs. Defining which proteins have been repurposed, which are involved in endocytosis and which proteins have differing stage-specific roles during parasite blood stage development will also contribute to a better understanding of the general vesicle trafficking system of malaria parasites.

## Materials and methods

### Cloning and plasmids used

Plasmids were generated using Gibson assembly [66] or T4 ligation using pARL1 based plasmids [67] and pSLI plasmids [30] (see S1 File for sequence of plasmids used). The GRASP and truncated STEVOR-SDEL (for ER) constructs were previously used [3]. The GFP-fused P40X reporter plasmid [42] was a kind gift from Jude Przyborski.

### Cell culture of *P. falciparum*

*P. falciparum* parasites (strain 3D7) [68] were cultured in 0+ erythrocyte (transfusion blood, Universität Klinikum Eppendorf, Hamburg) in RPMI1640 complete medium containing 0.5% Albumax (Life Technologies), 20 mM glucose and 200 mM hypoxanthine at 37˚C according to standard methods with a hematocrit of 5% in a microaerophilic atmosphere (1% $O_2$, 5% $CO_2$, and 94% $N_2$) [69].

### Parasite transfection

For transfection of *P. falciparum* parasites, mature schizonts were enriched by density separation through 60% Percoll ($2,000 \times g$, 8 min) and resuspended in transfection buffer containing 50 µg of plasmid DNA and electroporated using Nucleofector II (AAD-1001N, program U-033). Drug concentrations for the selection of transfected parasites were 4 nM for WR99210 (Jacobus Pharmaceuticals), 2 mg/ml for Blasticidin S (Invitrogen), 0.9 mM for DSM1 (MRA/BEI Resources), and 400 mg/ml for G418 (Sigma), as appropriate for the resistance marker of the respective plasmid. For the selection of integrants by SLI [30], a culture (parasitemia of 5% to 8%) was grown in the presence of G418 or DSM1 (selecting drug according to the SLI resistance marker on the plasmid). After the first detection of parasites by microscopic examination of Giemsa smears, verification of the desired genome integration was performed by PCR on genomic DNA prepared using QIAamp DNA Mini kit (see S1 Table for primers used).

### Preloading of red blood cells and invasion of parasites

Preloaded RBCs were prepared according to previously established protocols [11,13]. Two hundred microliter of RBC concentrate was washed in cold DPBS ($2,000 \times g$, 1.5 min) and 32 µl of packed RBCs were added to a lysis solution consisting of 64 µl lysis buffer (5 mM K2HPO4/20 mM D-Glucose (pH 7.4)), 1 µl 30 mM DTT, 2 µl 50 mM MgATP, and 1 µl (50 mg/ml) fluorescently labeled dextran (Alexa Fluor 647-conjugated 10 kDa dextran, Nanocs).

The resulting mixture was placed on ice and rotated overhead at 4˚C for 10 min. Resealing of the lysed RBCs was performed by gently adding 25 μl of 5× resealing buffer (750 mM NaCl/25 mM Na2HPO4 (pH 7.4)) to the mixture, followed by careful shaking (350 rpm, Eppendorf ThermoMixer F1.5) at 37˚C for 60 min. The preloaded cells were washed 3 times with RPMI and stored in RPMI at 4˚C. For the invasion of preloaded RBCs, trophozoite stage parasites were isolated from a mixed culture using Mini-Percoll [70,71]. After the harvesting and washing of the trophozoite layer, the parasites were mixed with the preloaded cells in a 2-ml Petri dish and cultivated for 43 h until the parasites had reinvaded the preloaded cells and developed to mid-trophozoite stages. Before imaging, the dish was split into two 1-ml Petri dishes and one was treated with rapalog (250 nM, Clontech) for 4 h to induce the knock-sideways while the other served as control.

## Hemoglobin immunofluorescence assay

To detect hemoglobin-filled vesicles, immunofluorescence assays (IFAs) were performed. Four ml of synchronized parasite culture (16 to 34 h post invasion) of the respective knock-sideway cell lines were divided into two 2-ml Petri dishes, one of which was grown with 250 nM rapalog for 6 h to induce the knock-sideways, while the other dish served as control without induction. Cells were washed with DPBS and applied on a ConA-coated (prepared as described [72]) 10-well slide (Thermo Scientific). After 10 min of incubation, excess cells were washed off with DPBS. Thereafter, 0.03% saponin in 1× PBS was added for 5 min to the wells on the slide to remove the host cell cytosol followed by 3 wash steps in 1× PBS. The cells were fixed at room temperature for 30 min in 1× PBS containing 4% formaldehyde, washed 3 times in 1× PBS followed by permeabilization with 0.1% Triton X-100 in 1× PBS for 10 min at room temperature. After 3 washing steps, the parasites were incubated in "blocking solution" (3% BSA in 1× PBS and 100 mg/ml ampicillin) for 1 h at room temperature. Thereafter, the cells were incubated at 4˚C overnight in blocking solution containing the primary antibody (Rabbit anti-hemoglobin (SIGMA, Cat. No. H4890), diluted 1/1,000). The parasites were washed 3 times for 5 min in 1× PBS before the second antibody (Alexa Fluor 647-conjugated goat anti-Rabbit antibody (Life Technologies)), diluted 1/2,000 in blocking solutions with 1 μg/ml DAPI (4′,6′-diamidine-2′-phenylindole dihydrochloride), was added and incubated for 1 h at room temperature. The cells on the slide were washed 3 times with 1× PBS, and the slide was covered with a coverslip using Dako fluorescence mounting medium for imaging. Analysis of the images was performed blinded to the condition of the respective sample.

## Co-immunoprecipitation (CoIP)

For CoIP experiments with endogenously expressed GFP-tagged PfRbsn5L, 50 ml of parasite culture (parasite lines Rbsn5L-2xFKBP-GFP-2xFKPBP$^{endo}$ + Rab5b-mCh$^{epi}$ or Rbsn5L-2xFKBP-GFP-2xFKPBP$^{endo}$ + VPS45-mCh$^{epi}$) was centrifuged (2,000 × g, 3 min), washed 3 times in DPBS, the RBCs lysed in 10 pellet volumes of 0.03% saponin in PBS, and washed 3 times with DPBS. The purified parasites were lysed in 200 μl RIPA buffer (150 mM NaCl, 10 mM TrisHCl (pH 7.5), 0.1% SDS, 1% TX100 containing 2× protease inhibitor cocktail (Roche), and 1 mM PMSF) and frozen at −80˚C. The lysate was thawed and cleared by centrifugation at 16,000 × g at 4˚C for 10 min and the supernatant was diluted with 70 μl of dilution buffer (150 mM NaCl10 mM TrisHCl (pH 7.5) containing 2× protease inhibitor cocktail and 1 mM PMSF). A sample of 50 μl was removed and mixed with 4× SDS sample buffer (IP-input extract). Fifteen μl of GFP-agarose bead slurry (ChromoTek GFP-Trap), equilibrated in dilution buffer, was transferred to the diluted supernatant and the tube rolled overhead overnight at 4˚C. The agarose beads were pelleted (2,500 × g, 3 min, 4˚C) and 50 μl of the supernatant

was transferred to a tube containing 4× SDS sample buffer (total extract after IP). The beads were washed 5 times with dilution buffer (centrifugation steps at 2,500 × g), and 50 μl of the supernatant from the last wash step was transferred to a tube containing 4× SDS sample buffer (last wash). The agarose pellet was incubated with 50 μl of 2× SDS sample buffer at 95°C for 5 min to elute proteins bound to the agarose beads. The beads were pelleted by centrifugation at 2,500 × g for 1 min and 50 μl of the supernatant was transferred to a fresh tube (eluate). Equivalent volumes of the IP-input, after IP lysate, last wash, and eluate were separated by SDS-PAGE for western blot analysis.

## SDS-PAGE and immunoblotting

CoIP samples were separated by SDS-polyacrylamide gel electrophoresis (PAGE), transferred to Amersham Protan membranes (GE Healthcare) in a tankblot device (Bio-Rad) using transfer buffer (0.192 M 848 Glycine, 0.1% SDS, 25 mM Tris-HCl (pH 8.0)) with 20% methanol. Blocking of the membranes (1 h) was done in 5% skim milk in PBS. Primary antibodies were diluted in PBS containing 5% skim milk and incubated with membranes overnight. Primary antibodies were: monoclonal mouse anti-GFP (Roche) (1:1,000), polyclonal rat anti-mCherry (Chromotek) (1:1,500), or rabbit anti-BIP [73] (1:2,000). After 3 wash steps in DPBS, the respective secondary antibody anti-mouse-HRP (Dianova) (1:3,000), anti-rat-HRP (Dianova) (1:2,500), or α-rabbit-HRP (Dianova) was added and incubated for 1 h at room temperature in PBS containing 5% skim milk. After washing the membranes 3 times with PBS, detection was done using enhanced chemiluminescence (Bio-Rad/Thermo Fisher), and signals were recorded using a ChemiDoc XRS imaging system (Bio-Rad).

## Fluorescence microscopy

Fluorescence microscopy was done essentially as described [74] using a Zeiss Axioscope M1 or M2 fluorescence microscope equipped with a LQ-HXP 120 light source. To stain the nuclei, DAPI was added to an aliquot of the culture at a final concentration of 1 μg/ml and incubated for 5 min at room temperature. Approximately 5 to 10 μl of the parasite cell suspension was transferred to a glass slide and covered with a coverslip. Images were acquired using a 63×/1.4 or 100×/1.4 oil immersion lens, a Hamamatsu Orca C4742-95 camera, and the Zeiss Axio Vision software. Different filter cubes (Zeiss cubes 44, 49, 64, and 50) corresponding to the fluorescent marker were used for excitation and detection. Images were processed using Corel PHOTO-PAINT X6 and arranged in CorelDraw X6. Line plots were generated in ImageJ [75] and graphs were generated in Excel.

Confocal imaging was done with an Evident Fluoview 3000 confocal microscope equipped with an Olympus 63×/1.5 oil immersion lens and 405 nm, 488 nm, and 561 nm laser lines. Image stacks were analyzed in Imaris 6.3, the images were recovered using the snapshot function and cropped, overlayed, and arranged using Corel PhotoPaint and CorelDraw v24.3.

## Transmission electron microscopy

Transmission electron microscopy was done as described [13]. Parasites 8 h after induction of the knock-sideways (with 250 nM rapalog) and the control cells grown without rapalog were Percoll-enriched and fixed with 2.5% glutaraldehyde (Electron Microscopy Sciences) in 50 mM cacodylate buffer (pH 7.4) for 1 h at room temperature. In some samples, the parasites were treated with saponin (0.03% in DPBS) for 15 min on ice followed by 3 times washing in DPBS to remove the cytosol from the host cell prior to the first fixation step. For postfixation and staining of the cell membranes, 2% osmium tetroxide (OsO4) in $dH_2O$ was added and incubated for 40 min on ice in the dark. After 3 washes with $dH_2O$, uranyl acetate (Agar

Scientific) was added for 30 min at room temperature. The cells were washed 2 to 3 times with dH$_2$0 and dehydrated in an ethanol series (50%, 70%, 90% (2×), 95% (3×), and 100%) for 5 min each. An epon-ethanol mixture (1:1) was added and incubated overnight at room temperature by shaking. The next day, cells were incubated with 100% epon (Carl Roth GmbH & Co. KG) for 6 h and then replaced with fresh epon. For polymerization, the sample was kept at 60˚C for 1 to 3 days. Samples were cut into 60 nm sections with an Ultracut UC7 (Leica) and examined with a Tecnai Spirit transmission electron microscope (FEI), equipped with a LaB6 filament and operated at an acceleration voltage of 80 kV. The EM image.emi/.ser-files were converted to 8-bit TIFF files using the TIA Reader Plugin for ImageJ [75]. False coloring was added using Corel PHOTO-PAINT X6.

### Parasite growth assay by flow cytometry

To monitor the growth of *P. falciparum* parasite cultures, flow cytometry was used to measure the parasitemia [30,76]. For this purpose, a culture with an adjusted parasitemia of 0.05% was divided into two 2-ml Petri dishes, one of which was grown with 250 nM rapalog to induce the knock-sideways of the protein of interest while the other was used as un-induced control without rapalog. The medium was changed daily (and rapalog replenished in the induced culture) and the parasitemia was measured once every 24 h for 5 consecutive days (2.5 growth cycles). To do the flow cytometry measurement, the culture was thoroughly mixed and 20 μl of each culture was added to 80 μl medium containing 1 μl DHE (0.5 mg/ml) and 1 μl Hoechst 33342 (0.45 mg/ml). After staining in the dark for 20 min, the parasites were inactivated by adding 400 μl 0.000325% glutaraldehyde in RPMI. Parasitemia was measured by flow cytometry using an LSR-II cytometer by counting 100,000 events using the FACSDiva software (BD Biosciences) as described [30,76]. Results were displayed using GraphPad Prism (version 7.04).

### Growth assays with synchronous parasites

For induction during the ring stage, (done with Rbsn5L-2x2FKBP-GFP[endo] + NLS[epi] and Rab5b-2xFKBP-GFP endo + NLS[epi] parasites) the parasites were synchronized with 5% sorbitol, split into 2 dishes, of which one received 250 nM rapalog while the other served as control and the parasites cultured for 10 h. Then, the sorbitol treatment was repeated to obtain parasites 10 to 18 h post invasion of which the rapalog culture corresponded to ring stages that had been on rapalog starting with a 0- to 8-h stage window. The parasites were then cultured at 37˚C and Giemsa smears were prepared twice a day at the time points indicated in the specific experiments. For each time point and condition at least 100 parasites were counted and the parasitemia and stages were recorded. For selected time points, DIC images were taken to assess the vesicle phenotype.

For the induction at the beginning of the trophozoite stage (done for Rbsn5L-2x2FKBP-GFP[endo] + NLS[epi] parasites), mature schizonts were purified from asynchronous cultures using a Percoll gradient, the parasites allowed to invade fresh RBCs at 37˚C for 8 h after which they were synchronized with 5% sorbitol to obtain 0- to 8-h ring stages. After culturing for 16 h (resulting in parasites of 16 to 24 h post invasion stage window), the culture was split and one received rapalog to 250 nM. Continued culturing, Giemsa smears, and DIC images were done as described for the ring-stage induction.

### Vesicle accumulation assay

Vesicle accumulation assays were performed according to previously established protocols [13]. A mixed culture containing up to 5% ring-stage parasites was synchronized by sorbitol (5%) treatment to obtain parasites 0 to 18 h post invasion, which subsequently were cultured

for 16 h to obtain trophozoite stage parasites (16 to 34 h post invasion). The culture was split into two 2-ml dishes, and one dish was treated with rapalog to a final concentration of 250 nM. After 0, 2, 4, 6, and 8 h of incubation at 37˚C, samples were collected, stained with DAPI, and imaged immediately. For vesicle assays performed with the double knock-sideways parasites, more tightly synchronized parasites were used and generated essentially as described [74]. Mature schizonts were purified from an asynchronous parasite culture using a Percoll gradient and were allowed to invade fresh RBCs at 37˚C for 3 h after which they were synchronized with 5% sorbitol to obtain 0- to 3-h old ring stages. Knock-sideways was induced 24 to 27 h post invasion by adding rapalog for 6 h before imaging. Numbers of vesicles accumulating in the trophozoite stage of the parasite (parasites with >3 nuclei were excluded) were counted blind to the conditions in the respective DIC images.

### Bloated food vacuole assay

Bloated food vacuole assays were performed as described [13]. The respective mixed parasite culture was synchronized twice with 5% sorbitol at 10 h intervals to obtain a ring stage parasite culture with a 10 to 18 h post invasion stage window. The parasites were cultured for 8 h (resulting in a stage window of 18 to 26 h post invasion) and divided into two 1-ml dishes to which E64 protease inhibitor (Sigma Aldrich) was added to a final concentration of 33 µm. One dish was additionally treated with rapalog (250 nM), while the other served as control. The parasites were cultured for 8 h, stained with 4.5 µg/ml DHE for 20 min at room temperature, washed once in RPMI, and imaged. The DIC image was used for counting the number of cells with bloated and non-bloated food vacuoles and measuring the parasite diameter. Analysis of the images was performed blinded to the condition of the respective sample.

### Vesicle assay using Cytochalasin D and KIC7 inactivation

A mixed culture of PfKIC7$^{endo}$ + 1xNLS parasites [3] was synchronized by two 5% sorbitol treatments 10 h apart to obtain parasites with an age of 10 to 18 h post invasion. These parasites were cultured for a further 14 h to obtain trophozoite stage parasites (24 to 32 h post invasion), the culture split into four 2-ml dishes of which in two the KIC7 knock-sideways was induced by addition of rapalog to a final concentration of 250 nM while the other 2 dishes were not treated. After 1 h, 1 dish without rapalog and 1 dish with rapalog were treated with Cytochalasin D to a final concentration of 10 µm. The parasites were cultured for additional 5 h before imaging by DIC. Numbers of vesicles accumulating in the parasites were counted blind to the conditions the respective DIC image originated from.

### Statistics

Statistical analyses were done using GraphPad Prism (version 7.04). If not stated otherwise, two-tailed unpaired $t$ tests were used and means, error bars (SD), and $n$ are given and indicated in the figure legends. The statistical methods used are specified in the figure legends. Cultures grown with and without rapalog were analyzed side by side and in the same experiment originated from the same parent culture split into 2 upon rapalog-addition. For microscopy, images were taken from randomly selected areas based on the DIC view. For vesicle accumulation and bloated food vacuole assays, blinding was done to obscure the nature of the sample. However, it should be noted that in most cases, this was not sufficient to obscure rapalog-treated from control cells as lack of bloating and vesicles in the parasite were very obvious.

## Supporting information

**S1 Fig. Domain motifs in PfRbsn5L. (A, B)** Multiple sequence alignment of (A) FYVE and (B) Rab-binding domains of the indicated proteins and regions. Residues indicated as conserved were colored according to their physical properties. Sequence logo of the conserved residues in A was generated from the sequences of all FYVE domains found in *H. sapiens*, *S. cerevisiae*, *T. brucei*, *A. thaliana*, *G. intestinalis*, *P. falciparum*, and *T. gondii*. Red asterisks show amino acids important for PI3P binding or specificity that are not conserved in the FYVE domains of PfRbsn5L and human Protrudin (which contains an FYVE domain not binding PI3P but other phosphoinositides) while they are conserved in PF3D7_1460100 (FCP), the likely PfEEA1 [36]. In B, similarity according to clustal omega (dots and asterisks) are indicated below the alignment; residues involved in hydrogen bonds that are conserved between both Rab-binding domains of human Rbsn5, according to PDB 1z0k and 1z0j, were marked with + above. **(C)** Structural alignment of AlphaFold2 predicted structure of PfRbsn5L FYVE domain compared to the experimental structures of human EEA1 and human RUFY1. **(D)** Expression of a tandem of the PfRbsn5L FYVE (2xFYVE) domain fused to mCherry in parasites expressing P40X-GFP (P40X) to mark PI3P positive regions. Nuclei were stained with DAPI; DIC, differential interference contrast; merge: overlay of green and red channels; size bar: 5 μm.
(PDF)

**S2 Fig. Confirmation of correct genomic integration and additional flow cytometry growth curves. (A)** Agarose gels showing PCR-products to assess correct integration of the SLI-plasmids into the genome of *P. falciparum* (3D7) parasites to obtain the indicated cell lines (see also S1 Raw Images). 3D7, parent cell line; INT, integration cell line; 5′, PCR product across the 5′ integration junction; 3′, PCR product across the 3′ integration junction; ori, original locus (absence showing lack of parasites with unmodified locus). M, marker with selected fragments indicated in bp. Primers used for integration confirmation and expected sizes of PCR products listed in S1 Table and sequences of the plasmids in S1 File. **(B)** Intensity profiles along the indicated lines in images from Fig 1C generated with ImageJ. Lines were drawn in both images using the synchronize images command and the plot profile values were used to draw graphs in Excel. Arrows show green foci and are color coded as in Fig 1C. **(C)** Live-cell microscopy images of PfRbsn5L-2xFKBP-GFP-2xFKBP$^{endo}$ parasites, co-expressing the ER-marker STEVOR-SP-mScarlet-SDL$^{epi}$ (SP-SDEL$^{epi}$). White arrows show PfRbsn5L$^{endo}$ foci close to the food vacuole that were added at the same position in the DIC/Hoechst image as green arrows to illustrate the position relative to the food vacuole. **(D)** Replicates of the flow cytometry growth curves shown in the main figures (cell lines indicated). The data underlying the growth curve replicates can be found in S1 Data under the main figure designation of the respective growth curve.
(PDF)

**S3 Fig. Inactivation of PfRbsn5L in synchronous parasites. (A, B)** Synchronous parasites were inactivated in rings (A) or at the start of the trophozoite stage (B) (see Materials and methods) and the development monitored based on Giemsa smears. The rings monitored correspond to parasites where PfRbsn5L was inactivated 0–8 h post invasion (hpi), the trophozoites at 16–24 hpi ($n$ = 2 independent experiments; Giemsa smears shown only for replicate 1). Note that cells scored as rings after prolonged growth arrest in + rapa in the ring induction showed various morphological alterations not specifically scored. The data underlying this figure can be found in S1 Data.
(PDF)

**S4 Fig. Additional data for bloated food vacuole and vesicle accumulation assays and vesicle accumulation phenotype and co-localization of PfRbsn5L with regions containing PI3-P. (A, B)** Superplots showing diameter in μm of parasites analyzed in the vesicle accumulation assay in Fig 1G (A) and of parasites analyzed in the bloated FV assay in Fig 2B (B). Parasites from $n = 3$ independent experiments are distinguished by blue, yellow, and black dots; two-tailed unpaired $t$ test of the means, $p$-values indicated; mean (red bar); error bars (black) show SD. **(C)** DIC example images showing the phenotype of time points of the parasites from the stage growth assay in S3 Fig that resulted in the growth phenotype (time points indicated). **(D)** Confocal microscopy images of PfRbsn5L$^{endo}$ parasites (PfRbsn5L: green channel) co-expressing the mScarlet tagged PI3P marker P40X (red channel) used for the experiments shown in Fig 2E. Top, single z-slice, bottom panels show 4 consecutive confocal z-slices (z1-4). White arrows: PI3P positive regions overlapping with PfRbsn5L accumulations adjacent or at the food vacuole. DIC, differential interference contrast; merge, overlay of red and green channels. Size bars, 5 μm. The data underlying this figure can be found in S1 Data under the main figure indicated for each figure part.
(PDF)

**S5 Fig. Additional Rab5b localization data and stage-specific growth phenotype. (A, B)** Live-cell microscopy images of PfRab5b$^{endo}$ parasites co-expressing the ER marker STEVOR-SP-mScarlet-SDL$^{epi}$ (SP-SDEL$^{epi}$) (A), and of PfRab5b$^{endo}$ schizonts (B) showing a location suggestive of the IMC. Merge, overlay of green and red channels. Arrows in A show: blue, ER membrane; white, parasite plasma membrane; orange, Rab5b foci at the FV not overlapping with the ER marker; purple, possible FV loop-like structure. Size bar 5 μm. Nuclei were stained with DAPI. **(C)** Growth assay after conditional inactivation of Rab5b in synchronous rings monitoring progression through the cycle based on Giemsa smears (the parasites monitored correspond to parasites where Rab5b was inactivated 0–8 h post invasion (hpi)) ($n = 2$ independent experiments). Example Giemsa smear images shown for replicate 1. **(D)** DIC example images showing the phenotype in the parasites from the stage growth assay in C of the time points (indicated) leading to the growth phenotype. The data underlying this figure can be found in S1 Data.
(PDF)

**S6 Fig. PfRbsn5L and PfVPS45 co-localize with PfRab5b positive membranes. (A–C)** Live-cell microscopy images of the indicated stages of PfRbsn5L-2xFKBP-GFP-2xFKBP$^{endo}$ parasites, co-expressing PfVPS45-mCh$^{epi}$ (A), PfRbsn5L-2xFKBP-GFP-2xFKBP$^{endo}$ parasites, co-expressing PfRab5b-mCh$^{epi}$ (B), and PfVPS45-2xFKBP-GFP$^{endo}$ parasites, co-expressing PfRab5b-mCh$^{epi}$ (C). DIC, differential interference contrast; endo, endogenous; epi, episomal. Scale bar, 5 μm. Nuclei were stained with DAPI. **(D–F)** Confocal microscopy images of PfRbsn5L$^{endo}$ + Rab5b$^{epi}$ parasites (PfRbsn5L: green channel; Rab5b: red channel) showing 4 consecutive confocal z-slices (z1-4) per cell. White arrows: Regions where PfRbsn5L overlaps with Rab5b. Nuclei were stained with Hoechst; DIC, differential interference contrast; merge, overlay of red and green channels. Size bars, 5 μm.
(PDF)

**S7 Fig. PfVPS45 interacts with PfRbsn5L.** Replicas and complete blots of 3 independent experiments of Immunoprecipitation (IP) of extracts from PfRbsn5L$^{endo}$ parasites co-expressing PfVPS45$^{epi}$ (one cropped version is shown in Fig 5D). Note that only the indicated band on the anti-BIP blot in experiment 2 corresponds to BIP, the other bands are from a previous probing of the same membrane. IIP, IP-input extract; UB, unbound (total extract after IP); W, last wash; E, eluate; endo, endogenous; epi, episomal. See S1 Raw Images for uncropped blots

merged with marker as obtained directly from the ChemiDoc XRS imaging system.
(PDF)

**S8 Fig. PfRab5b interacts with PfRbsn5L.** Replicas and complete blots of 3 independent experiments of Immunoprecipitation (IP) of PfRbsn5L$^{endo}$ parasites co-expressing PfRab5b$^{epi}$ (one cropped version is shown in Fig 5E). Note that only the indicated band on the anti-mCherry and anti-BIP blots in experiment 3 correspond to the respective proteins, the other bands are from a previous probing of the same membrane. IIP, IP-input extract; UB, unbound (total extract after IP); W, last wash; W1, first wash; W2, second wash; E, eluate; endo, endogenous; epi, episomal. See S1 Raw Images for uncropped blots merged with marker as obtained directly from the ChemiDoc XRS imaging system.
(PDF)

**S9 Fig. Modified SLI-sandwich plasmid enabling endogenous tagging and mislocalization of POIs and double conditional inactivation strategy. (A)** Schematic of selection linked integration strategy combined with a mislocalization cassette on the same plasmid. **(B)** Scheme of a selection linked integration strategy enabling endogenously tagging and simultaneous mislocalization of 2 POIs in the same parasites. **(C)** Schematic illustration of simultaneous double mislocalization of KIC7 (which inhibits endocytosis at the PPM in an early step) and PfRbsn5L (which inhibits transport of HCC-filled vesicles to the food vacuole), showing the expected reduction in vesicle accumulation compared to PfRbsn5L inactivation alone if the 2 processes are serially linked.
(PDF)

**S1 Table. Primers used for assessing correct genomic integration of plasmids.**
(DOCX)

**S1 File. Plasmids generated for this work.**
(DOCX)

**S1 Data. All numerical values underlying this study.**
(XLSX)

**S1 Raw Images. Raw images of S2A Fig, Figs 5D and S7, Figs 5E and S8.**
(PDF)

# Acknowledgments

We are grateful to Jacobus Pharmaceuticals for providing WR99210. DSM1 (MRA-1161) was received from MR4/BEI Resources, NIAID, NIH. We thank Tim-Wolf Gilberger for providing the anti-BIP antibodies. We thank Isabelle Henshall and Stephan Wichers-Misterek for their comments on the manuscript.

# Author Contributions

**Conceptualization:** Tobias Spielmann.

**Formal analysis:** Ricarda Sabitzki, Anna-Lena Roßmann, Andrés Guillén-Samander, Hannah Michaela Behrens, Tobias Spielmann.

**Funding acquisition:** Tobias Spielmann.

**Investigation:** Ricarda Sabitzki, Anna-Lena Roßmann, Marius Schmitt, Sven Flemming, Andrés Guillén-Samander, Ernst Jonscher, Katharina Höhn, Ulrike Fröhlke.

**Methodology:** Ricarda Sabitzki.

**Project administration:** Tobias Spielmann.

**Resources:** Ricarda Sabitzki, Sven Flemming, Andrés Guillén-Samander, Ernst Jonscher, Tobias Spielmann.

**Supervision:** Tobias Spielmann.

**Visualization:** Ricarda Sabitzki, Anna-Lena Roßmann, Hannah Michaela Behrens, Tobias Spielmann.

**Writing – original draft:** Ricarda Sabitzki, Tobias Spielmann.

**Writing – review & editing:** Ricarda Sabitzki, Anna-Lena Roßmann, Marius Schmitt, Sven Flemming, Andrés Guillén-Samander, Hannah Michaela Behrens, Ernst Jonscher, Katharina Höhn, Ulrike Fröhlke, Tobias Spielmann.

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
