## [Editor Report · Decision Letter 0]

5 May 2023

Dear Dr. Spielmann, 

Please allow me to first apologize for the delay in the processing of your manuscript. Thank you for submitting your manuscript entitled "Identification of a Rabenosyn-5 like protein and Rab5b in host cell cytosol uptake reveals conservation of endosomal transport in malaria parasites" for consideration as a Research Article by PLOS Biology.

Your manuscript has now been evaluated by the PLOS Biology editorial staff, as well as by an academic editor with relevant expertise, and I am writing to let you know that we would like to send your submission out for external peer review.

Once your full submission is complete, your paper will undergo a series of checks in preparation for peer review. After your manuscript has passed the checks it will be sent out for review. To provide the metadata for your submission, please Login to Editorial Manager (https://www.editorialmanager.com/pbiology) within two working days, i.e. by May 07 2023 11:59PM.

Kind regards,

Paula

---

Senior Editor

PLOS Biology

---

## [Decision Letter · Decision Letter 1]

19 Jun 2023

Dear Dr. Spielmann,

Thank you for your patience while your manuscript "Identification of a Rabenosyn-5 like protein and Rab5b in host cell cytosol uptake reveals conservation of endosomal transport in malaria parasites" was peer-reviewed at PLOS Biology. It has now been evaluated by the PLOS Biology editors, an Academic Editor with relevant expertise, and by several independent reviewers. 

In light of the reviews, which you will find at the end of this email, we would like to invite you to revise the work to thoroughly address the reviewers' reports.

As you will see below, the reviewers agree that the live fluorescence imaging is not of sufficient quality to justify the conclusions. In particular, we consider that it is important that you address the concerns regarding PI3P and rbsn5 co-localisation and the lack of an ER marker. Reviewer #2 and #3 agree that Figures 2E and F require some correlation for P40 and rbsn5. We consider that given the high rbsn5 background staining this will be difficult with existing images. We think that the authors indirect evidence of causation between rbsn5 KS and increased P40 staining vesicles is adequate to implicate rbsn5 in early endosomes with PI3P. We agree with reviewer #1 that the host cell cytosol in vesicles accumulating after PfRbsn5 KS in fig 2G is adequate evidence that the vesicles are derived from host cell endocytosis and we think that doing further experiments to prove the origin of the vesicles is not necessary. We also agree that the vacuole bloating assay shows bloating in E64 treated parasites unless rbsn5 is KS, therefore we think that other comparisons are not strictly needed. We consider that it is not needed for you to remove results sections, eg KIC7, you can decide what parts of the story you are telling are important. Please address the rest of the reviewers' issues. 

Given the extent of revision needed, we cannot make a decision about publication until we have seen the revised manuscript and your response to the reviewers' comments. Your revised manuscript is likely to be sent for further evaluation by all or a subset of the reviewers.

**IMPORTANT - SUBMITTING YOUR REVISION**

*Re-submission Checklist*

*Published Peer Review*

*PLOS Data Policy*

*Blot and Gel Data Policy*

Sincerely,

Paula

---

Senior Editor

PLOS Biology

REVIEWS:

Reviewer #1: Secretory and endocytic trafficking in malaria parasites.

Reviewer #2: Host cell-parasite interactions.

Reviewer #3: Cell signalling.

Reviewer #4: Malaria parasites biology.

Reviewer #1: In this manuscript, the authors convincingly demonstrate that a Plasmodium falciparum Rabenosyn-5 like protein and Rab5B are required for the delivery of hemoglobin containing vesicles to the food vacuole. They further show that the 2 proteins form a complex with PfVPS45, previously also shown by this group of investigators to be involved in the same process. Finally, they provide evidence using a very clever double conditional knock sideways assay that the PfKelch 13 compartment is upstream of the identified Rbsn5/VPS45/Rab5 complex. 

This study is of broad interest for the readership of Plos Biology as it demonstrates that the pathway of host cell-cytosol uptake in the malaria parasite contains both parasite specific adaptations and features of a canonical endosomal system. The work is of high quality, novel and the paper is well written. All the data is provided in the manuscript and in the sup material. 

Major comments:

1. -An important aspect that is not addressed much in the manuscript is the fact PfRbsn5 is likely a phosphoinositide binding protein due to the presence of a FYVE domain.

-Are the conserved PI3P binding attributes present in PfRbsn5? It would be important to show an alignment of the domain with well characterized FYVE domains, highlighting the conserved features.

-It would be important for the authors to show whether the FYVE domain of PfRbsn5 actually binds PI3P. This could be done with a recombinantly expressed domain of WT and mutant domains and liposome binding assays. At the very least, the authors could episomally express in parasites a WT and a mutant of the PfRbsn5 FYVE domain and see whether mutations of the putative PI3P binding residues abrogate the localization.

 -If PfRbsn5 does bind PI3P, the authors should discuss where they think this occurs in the pathway. Does PfRbsn5 located on HCC vesicles bind PI3P on the food vacuole membrane or does it bind PI3P on the HCC vesicles? In the canonical endosomal pathway, early endosomes are labelled with PI3P which is then transformed in PI(3,5)P2 during maturation to late endosomes/multivesicular bodies. It is really intriguing that in Pf, PI(3,5)P2 has not been detected and that the food vacuole is massively labelled with PI3P despite being at a later/final step in the endosomal pathway. This is very different from model organisms. As such, I think that writing "reveals conservation of endosomal transport in malaria parasites" is too strong since only some elements are conserved. To be clear, I am not suggesting the authors try to answer this question in the current manuscript but since they are one of the (if not the) labs that have made the most critical contributions to the molecular understanding of the HCCU pathway in P. falciparum they certainly are in a position to speculate. 

2. In Fig. 5A, whilst there clearly is some overlap between Rsbn5 and Rab5B, is it not 100%. In addition, in 5E, only a very small amount of Rab5B is pulled down by Rbsn. Furthermore, in 5H, Rab5B is not mislocalized in the Rnsb5 KS. This suggests that only a small fraction of total Rab5B complexes with Rbsn5. Can the authors comment on this? I'm not sure it is only a question of stability of the interaction as stated by the authors. This could be answered by crosslinking the parasites before performing the pulldown. If stability is the cause, then much more Rab5B should then be pulled down under these conditions.

Minor comments;

-In the title of Fig 3, I think it would be better to not use the HCCU abbreviation but instead use host-cell cytosol uptake.

Reviewer #2: In their manuscript, the authors have investigated the function of a rabenosyn 5 (RBsn5) homolog in Plasmodium falciparum (PfRbsn5) acting together with VPS45 and Rab5 in the endosomal vesicular transport of host hemoglobin to the food vacuole. Through gene/protein expression manipulation, they conclude that hemoglobin uptake involves these proteins, with some parasite specificities (Kelch) and conserved mechanisms for a process of endocytosed material into a cell -as it is known in other organisms that Rbsn5, VPS45 and Rab5b function in one complex. Overall, data are well-presented, sometimes overinterpreted but their innovation is somehow limited. Some controls are missing.

Figure 1A: the sequence of rabenosyn 5 (RBsn5) homolog in P. falciparum is shown but several differences in domains are noticeable with human or yeast RBsn5/Vac1, with such a low sequence conservation, makes the designation of this protein as PfRbsn5 perplexing. The choice of another name reflecting 'RBsn5-like protein' seems more appropriate.

Figure 1B: the food vacuole revealed by hemozoin content on DIC images does not contain PfRbsn5 but PfRbsn5 is in juxtaposed vesicles (purple arrow). The sentence: "a signal was present at the food vacuole in trophozoite stages (Figure 1B, dark blue arrow) with a more intense focus at the food vacuole in schizont stage parasites (Figure 1B, purple arrows)" is overstated. What is the authors' interpretation about PfRbsn5 signal detected in the nucleus (light blue arrow) at young blood stage in Fig. 1B and 1C- like the mislocalizer (nmd3'1xNLS-FRB-mChepi)?

Figure 1C: the source of fluorescently tagged Graspepi /plasmid is not indicated. Same for mScarlet tagged P40PX in Figure 2E.

Fig. 2A mirrors data I Fig. 1F but a clarification for the origin of these vesicles would be informative with an immunostaining with a PPM and PVM markers to assess their formation through endocytosis. Same comment for Figure 3.

Fig. 2B misses a control for food vacuole size of untreated, WT parasites to compared with Rbsn5-deficient parasites exposed to rapalog and E64. Same comment for Figure 3.

Fig. 2E is important to assess PI3P at the PfRbsn5 localization. To this reviewer, the signal for PI3P and Rbsn5 colocalization (yellow arrow) is not too convincing. Instead of showing the overlaid images on the DIC, it will be better to show the merge of the green and red signals with quantification (PDM+ values). On some images the green signal seems smaller that the red one. What is the meaning of the blue arrows relative to Rbsn5 mislocalization (which is located in the nucleus)? 

Figure 4D: the author must clarify the origin of the membrane pointed with the blue arrow to assess this is derived from the PVM. Or at least membrane pointed with the yellow arrow being the PPM as the parasite seems pretty messed up with high vacuolization upon all the treatments. For example, the membrane of the food vacuole is not apparent.

Figure 5B, F, I for the control conditions (no rapolog) are not convincing to assess Rab5 and Rbsn5 colocalization at this resolution, in contrast to VSP45 localization with either Rab5 or Rbsn5 more compelling.

Figure 6 about the cytostomal origin of the rbsn5 positive vesicles should be placed after Figure 4.

The Discussion section is more or less a summary of the Results and could be combined.

Reviewer #3: This work described in this manuscript by Sabitzki et al. attempts at understanding the role of small GTPase PfRab5b and its putative homologue of Rabenosyn 5 (PfRbsn5) in the malaria parasite. A reverse and forward genetics approach is taken for these studies, which involves the conditional inactivation of these two and other related proteins and these mutants are mainly used to evaluate the process of host erythrocyte cytosol-which is mainly composed of haemoglobin-uptake (HCCU). Previously, PI3K/VPS34 and VPS45 have been demonstrated to play a role in this process in malaria parasite. Given that Rab5 is known to be an effector of endocytic pathways regulated by VPS34 and VPS45 in yeast and mammals, the finding that PfRbsn5 and PfRab5b are also involved in HCCU were not surprising. I feel in general the study is nicely organized and experiments have been planned properly and executed effectively. However, at times authors have over-interpreted the data and need more experimental evidence to support some of the conclusions. 

Specific comments/queries:

1. Fig. 1A. The sequence comparison between PfRabsn5 with homologues indicated a lack of several key elements like C2H2 motif. It is important to provide the sequence comparison between the FYVE and Rab binding domains of various homologues. 

2. Given that PI3P is a key player in endocytic processes like HCCU and PfRabsn5 has a FYVE domain, it is extremely important to demonstrate that PfRabsn5 interacts with PI3P via its FYVE domain and its cellular localization is dependent on this interaction. These experiments are essential for this study.

3. Fig. 1C. Co-localization between Rbsn5 and Golgi marker GRASP is suggested. While some overlapping localization is observed, better images are needed to make this point. It is important to perform co-localization with an ER marker. Typically, Golgi is not known to contain much PI3P and is mainly rich in PI4P. 

4. Figure 1E. The inactivation of PfRbsn5 impairs asexual development of the parasite. It is important to find out which stage and the process of parasite development is regulated. A more detailed analysis of the mutant is needed to address this issue. 

5. Fig. 2B/C. It is important to indicate the stage and/or time after rapamycin addition at which parasites used for the experiments reported in this and other figures.

6. Figure 3B. PfRab5b is reported to be present at the food vacuole and also at the E

---

## [Decision Letter · Decision Letter 2]

17 Apr 2024

Dear Dr Spielmann,

Thank you for your patience while we considered your revised manuscript "Identification of a Rabenosyn-5 like protein and Rab5b in host cell cytosol uptake reveals conservation of endosomal transport in malaria parasites" for publication as a Research Article at PLOS Biology. This revised version of your manuscript has been evaluated by the PLOS Biology editors, the Academic Editor and the original reviewers.

Based on the reviews and on our Academic Editor's assessment of your revision, we are likely to accept this manuscript for publication, provided you satisfactorily address the remaining points raised by the reviewers and the following data and other policy-related requests.

IMPORTANT - please attend to the following:

a) Please change your title to the following: "Role of Rabenosyn-5 and Rab5b in host cell cytosol uptake reveals conservation of endosomal transport in malaria parasites"

b) Please attend to the remaining requests from reviewer #1. You'll see that reviewer #2 is satisfied.

c) I discussed the requests from reviewer #3 with the Academic Editor, who said "The authors have been suitably cautious in describing their findings but they should address the reservations of R3 by inclusion of text in the introduction clarifying that the binding specificity of PfRbsn5L remains unknown. Their transformation of parasites with recombinant FYVE domains that do not colocalise with PI3P doesn’t prove PfRbsn5L doesn’t bind PI3P but I accept that this proof is not trivial and could be itself an independent publication." Thus, while we understand that these experiments would greatly strengthen the study, addressing these concerns textually will be acceptable for publication at this stage.

d) Many thanks for providing the underlying data in S1 Data. Please cite the location of the data clearly in all relevant main and supplementary Figure legends, e.g. “The data underlying this Figure can be found in S1 Data.”

e) Please make any custom code available, either as a supplementary file or as part of your data deposition.

We expect to receive your revised manuscript within two weeks. 

*Published Peer Review History*

*Press*

Sincerely,

Roli Roberts

Roland Roberts, PhD

Senior Editor

rroberts@plos.org

PLOS Biology

CODE POLICY

Per journal policy, if you have generated any custom code during the curse of this investigation, please make it available without restrictions upon publication. Please ensure that the code is sufficiently well documented and reusable, and that your Data Statement in the Editorial Manager submission system accurately describes where your code can be found.

We require the original, uncropped and minimally adjusted images supporting all blot and gel results reported in an article's figures or Supporting Information files. We will require these files before a manuscript can be accepted so please prepare and upload them now. Please carefully read our guidelines for how to prepare and upload this data: https://journals.plos.org/plosbiology/s/figures#loc-blot-and-gel-reporting-requirements

DATA NOT SHOWN?

REVIEWERS' COMMENTS:

Reviewer #1:

In this revised version, the authors have done an excellent job with providing additional data to support their conclusions. They have also reworded some sentences to better highlight that Pf has conserved some aspects of a canonical endosomal system but that there are still important differences. I have only very minor comments and recommend publication of this high quality piece of work.

Comments:

1- -The absence of PI3P binding of the PfRbsn5L FYVE domain is a really interesting result and it is possible that, as suggested by the authors, it could bind other PIP species like the FYVE domain of Protrudin. I think it would be important to specify that the latter binds to PI(4,5)P2, PI(3,4)P2 and PI(3,4,5)P3 and comment on what potential PIP species the PfRbsn5L FYVE domain could bind to. Ebrahimzadeh et al (PMID: 29154995) had previously reported that overexpressing sensors specific to either PI(3,4)P2 or PI(3,4,5)P3 in P. falciparum blood stages resulted in a broad cytosolic signal, like what was seen here with the PfRbsn5L FYVE domain. This potentially suggest that it could bind either one of these PIPs. Since a PI(4,5)P2 sensor was shown to strongly label the parasite plasma membrane and potentially the cytostome (PMID: 29154995), it is unlikely that the PfRbsn5L FYVE domain would bind this particular species.

2-I have found a few instances in the text and figs where Rbsn5 was not changed to Rbsn5L. 

3-In the finale sentence of the introduction, authors write: 

"Overall, our data suggest that HCCU consists of a parasite-specific initial part at the PPM that delivers endocytosed material into a more canonical endosomal system."

I think it is important to specify that only some aspects are like the canonical endosomal system, several others are not, as discussed in the manuscript. It might seem like a trivial suggestion but I think it is critical for the readers to understand that not only the initial steps of HCC endocytosis contain parasite specific biology.

4- In the Discussion, the authors write:

"PfRab5bL has a role akin to that of Rab5 isoforms in other organisms"

I presume that they are talking about PfRab5b therefore the L should be deleted.

5- In the next sentence, the authors write:

"However, there are also important differences. We did not find any evidence for PI3P binding of its FYVE domain"

I think "its" should be replaced by "the PfRbsn5L" for clarity.

Reviewer #2:

This reviewer is globally satisfied by the authors improvements of this manuscript. The new confocal microscopy images provided have increased the plausibility of the findings.

Reviewer #3:

In this revised version of the manuscript by Sabitzki et al. , authors have tried to address issues raised by me and other reviewers. However, one of the major issues remains unresolved, which has also been raised by other reviewers: 

I feel it is extremely important to demonstrate that PfRabsn5 interacts with PI3P via its FYVE domain and its cellular localization is dependent on this interaction. Similar queries have been raised by other reviewers as well. Authors have stated reasons like "issues" related PIP-strips and non-"trivial" nature of liposome assays for not performing these important assays. If done with proper positive and negative controls experiments with PIP-strips can be very informative and continued to be used. The authors have relied on in silico analysis and proposed that PfRabsn5-FYVE domain may resemble me related to non-PI3P binding FYVE domains. It is important to know if PfRabs5 binding domain interacts with any other PIPs especially when comparisons are drawn to protruding, which interacts with PI3P with less affinity in comparison to PI(3,4)P2 and PI(3,4,5)P3 (Gil et al. JBC2012). They have relied on the overexpression of 2XFYVE domain construct to suggest that it is not targeted to PI3P rich locations. It is possible that PfRabsn5-FYVE binds with less affinity with PI3P or does not interact with it at all; either way it needs biochemical demonstration.

---

## [Editor Report · Decision Letter 3]

25 Apr 2024

Dear Dr Spielmann,

Thank you for the submission of your revised Research Article "Role of Rabenosyn-5 and Rab5b in host cell cytosol uptake reveals conservation of endosomal transport in malaria parasites" for publication in PLOS Biology. On behalf of my colleagues and the Academic Editor, Michael Duffy, I'm pleased to say that we can in principle accept your manuscript for publication, provided you address any remaining formatting and reporting issues. These will be detailed in an email you should receive within 2-3 business days from our colleagues in the journal operations team; no action is required from you until then. Please note that we will not be able to formally accept your manuscript and schedule it for publication until you have completed any requested changes.

Sincerely, 

Roli Roberts

Senior Editor

PLOS Biology

rroberts@plos.org